# KITINet: Kinetics Theory Inspired Network Architectures with PDE Simulation Approaches

## Abstract

Despite the widely recognized success of residual connections in modern neural networks, their design principles remain largely heuristic. This paper introduces KITINet (KInetics Theory Inspired Network), a way that reinterprets feature propagation through the lens of non-equilibrium particle dynamics and partial differential equation (PDE) simulation. We propose a new residual module that models feature updates as the stochastic evolution of a particle system, numerically simulated via a discretized solver for the Boltzmann transport equation (BTE). This formulation mimics particle collisions, enabling additional neuron-wise information propagation via physical interactions. Additionally, we reveal that this mechanism is an implicit regularization approach that induces network parameter condensation during training, where parameters progressively concentrate into a sparse subset of dominant channels. Experiments on large language modeling, image classification, scientific computation, and text classification show consistent improvements over classic network baselines, without additional inference cost.

## 1 Introduction

Residual connections have become a cornerstone of modern networks, enabling the training of exceptionally deep nets by alleviating vanishing gradients and stabilizing feature propagation. From ResNets [15] in vision to Transformers [43] in texts, residual mechanisms underpin state-of-the-art architectures. Recent advances have further explored residual learning through dynamical systems [7; 3], where iterative updates are analogized to differential equations. Concurrently, physics-inspired neural networks have gained traction, with frameworks such as PDE networks [26; 27] and Hamiltonian networks [42; 14] demonstrating that embedding physical principles into architectures can enhance physical interpretability and generalization. However, while these works highlight the potential of interdisciplinary design, the fusion of kinetic theory, particularly particle dynamics and collisional processes, with residual learning remains largely unexplored.

Despite their empirical success, existing residual modules are mostly designed heuristically. E.g., standard skip connections propagate features via simple additive operations, neglecting the rich dynamics of stochastical multi-particle interactions or energy exchange in non-equilibrium systems. As established in [40], the entropy-increasing behavior of feature representations is pervasive in architectures exhibiting information bottlenecks, such as GPT and ResNet. From a physical standpoint, particle collisions enhance macroscopic viscosity [12], easing the network's burden to produce external forces and yielding smoother force fields and more condense parameterization. From a mathematical perspective, stochastic collisions can be regarded as a source of implicit regularization.

However, existing dynamical systems perspectives reinterpret residual networks as discretized ODEs [7; 35; 34], failing to account for stochastic, collision-driven interactions that govern particle systems. This gap leaves critical questions unanswered: Can residual learning be reimagined through the lens of kinetic theory? How might collisional dynamics, as modeled by BTE, inform adaptive feature refinement? Prior physics-inspired architectures [39; 45] have not rigorously bridged particle-based simulation with parameter sparsity mechanisms, nor uncovered the phenomenon of network parameter condensation [46] i.e. training concentrates parameters into a sparse subset of channels, via a physics-grounded framework.

This paper introduces KITINet, a kinetics theory inspired network architecture that reformulates residual learning as a stochastic particle simulation governed by the BTE. We propose a novel residual module where feature updates emulate the collisional evolution of a multi-particle system: each

channel acts as a "particle" whose interactions are simulated via a discretized PDE solver, and adaptively redistributes information through physics-informed collision operators. This approach not only aligns feature propagation with non-equilibrium thermodynamics but also induces network parameter condensation, a phenomenon where gradients during training progressively sparsify parameters into dominant channels. Extensive experiments on language model pre-training, image classification, PDE operator learning, and text classification validate KITINet's efficacy, outperforming GPT2, ResNet, and BERT. By unifying kinetic theory with deep learning, it establishes a new paradigm for designing interpretable, physics-grounded architectures. **The highlights of the paper are:**

- This paper proposes a novel residual connection module to replace only the residual addition, which formulates the feature updating process as the evolution of a kinetic particle system and implements the module by simulating random particle collisions using a numerical algorithm of the BTE.
- It physically and mathematically promotes the recently heated phenomenon called network parameter condensation in training [46].
- Experimental results demonstrate that the proposed module achieves performance improvements over baseline models on language model pre-training, image and text and PDE tasks.
- It introduces a principled way to selectively embedding PDE structures into neural architectures.

## 2 PRELIMINARIES

### 2.1 KINETIC THEORY AND NUMERICAL ALGORITHM

The kinetic molecular theory of ideal gases is given as four postulates [25]:

1. A gas consists of particles called molecules, which are all alike in a given type of gas.
2. The molecules are in motion, and Newton's laws of motion may presumably be applied.
3. The molecules behave as elastic spheres with small diameters. Therefore, the space they occupy may be disregarded, and the collisions between them are energy-conservative.
4. No appreciable forces of attraction or repulsion are exerted by the molecules on each other.

When the particle system becomes too dense, it becomes necessary to describe the particle dynamics using distributions rather than trajectories. In kinetic theory, interaction sparsity is governed not by the absolute number of particles (e.g., the Avogadro constant), but by the ratio of the mean free path $\lambda$ to the characteristic system length $L$. This ratio—known as the Knudsen number $\text{Kn} = \lambda/L$. As detailed in Equation (4) later in the text, the effective mean free path is normalized to $\lambda = 1$, as defined in the collision modulation term $(U_r)_{i,j} = e^{-(X_r)_{i,j}}$. Due to commonly used normalization schemes (e.g., BatchNorm), the typical feature-space distance is approximately $L \approx 3.29$, corresponding to the 90th percentile of data spread. This yields a Knudsen number of $\text{Kn} = \lambda/L \approx 0.30$, which lies well within the valid kinetic regime for the BTE (commonly $0.01 < \text{Kn} < 10$).

The density function $f$ in the 7-dim phase space is defined as $dN = f(\boldsymbol{x}, \boldsymbol{p}, t)\, d^3\boldsymbol{x}\, d^3\boldsymbol{p}$. Assuming the displacement and momentum $\boldsymbol{x}, \boldsymbol{p}$ satisfy the Hamiltonian equations, and external force represented as $F_{ex}$, then $f$ satisfies the **Boltzmann transport equation (BTE)** [5]:

$$\frac{\partial f}{\partial t} + \frac{\boldsymbol{p}}{m} \cdot \nabla_{\boldsymbol{x}} f + F_{ex} \cdot \nabla_{\boldsymbol{p}} f = \left( \frac{\partial f}{\partial t} \right)_{coll} \tag{1}$$

where the right-hand side term describes the changes in the distribution due to particle collisions, which can only be approximated by an empirical formula. The BTE is a partial differential equation (PDE) that describes the evolution of the distribution function $f$ over time. There are various numerical methods to solve the BTE, such as the Direct Simulation Monte Carlo (DSMC) method [4] and the lattice Boltzmann method [21].

### 2.2 DIRECT SIMULATION MONTE CARLO (DSMC)

The DSMC [4] is a stochastic method that simulates the particle motion to solve BTE for dilute gas. Unlike molecular dynamics, each particle here represents $F_N$ molecules in the physical system. It divides the space into small cells and evolves the position and velocity of particles in each cell. The evolution consists of three steps: 1) Drift, 2) Wall Collision, 3) Particle Collision.

The first two steps are deterministic. The drift step moves the particles by assuming they move in straight lines without collision. The wall collision step checks if the particles collide with the wall and resets their velocity according to the boundary conditions.

---

**Algorithm 1** KITINet (with training and inference).

---

1: **Input:** Input $x \in \mathbb{R}^D$, residual $v \in \mathbb{R}^D$, hyper-parameters: dt, n_ divide, coll_coef;
2: **Output:** Output $x' \in \mathbb{R}^D$.
3: If model is in the inference mode, **Return** $x + dt * v$;
4: Reshape $x, v$ to collision_heads $\times N$ matrices $X, V$, where $N = D/$collision_heads;
5: Calculate relative properties $X_r, V_r$ and center-of-mass properties $X_{cm}, V_{cm}$ by Equation (2).
6: Calculate the full velocity change $\Delta V$ by Equation (3);
7: Select collision pairs by Equation (4);
8: Apply velocity and position change by Equation (5), get new position $X'$;
9: **Return** $x'$ flattened from $X'$;

---

The last step is stochastic. The particles are sorted into spatial cells, and only particle pairs in the same cell are selected to collide. The collision probability depends on the molecular interaction model. For a more detailed information about DSMC, please refer to Section C.

### 2.3 NETWORK PARAMETER CONDENSATION

Condensation of a neural network [50] describes the phenomenon where neurons in the same layer gradually form clusters with similar outputs during training. This process leads to the alignment or grouping of neurons that respond to related patterns in the input data. For evaluating parameter condensation, the cosine similarity is used as a natural and effective measure: $D(u, v) = \frac{u^\top v}{(u^\top u)^{1/2}(v^\top v)^{1/2}}$.

Extensive prior experimental phenomena and theoretical studies [48; 8] have established that the condensation phenomenon indicates when keeping the parameter within the same order of magnitude, the condensation phenomenon shows improvements in model generalization performance.

## 3 METHODOLOGY: THE KITINET ARCHITECTURE

As Figure 1 shows, we consider the network as the external force $F_{ex}$, and the hidden layer input is the position distribution of the particles $f$ in Equation (1). Each layer provides the velocity of the particle. During training, the DSMC-inspire module KITINet takes the residual connections $x$ and residuals $v$ as inputs, modeling the the remaining dynamics of Equation (1). It simulates the particle motion with collisions, permits particles to interact through pairwise encounters and to change their velocities, and outputs the position after a time step. In contrast, the structure in inference is the same as the vanilla network with KITINet turned off, simulating the particle motion without collisions and permitting particles to cross through each other without interacting or altering their velocities.

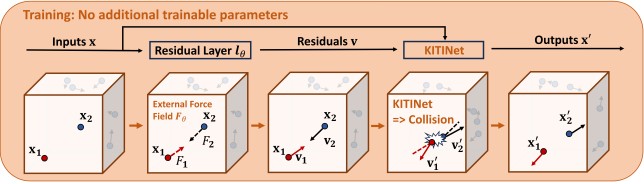

(a) KITINet training process

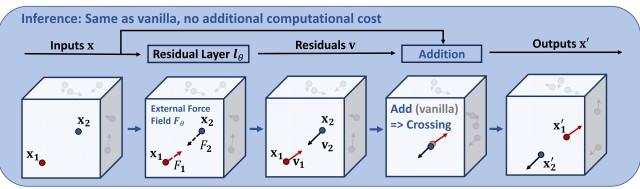

(b) KITINet inference process

Figure 1: In our design, only the residual addition is replaced by a trainable-parameter-free KITINet module during training, leaving inference unchanged. It gives a physical meaning: inputs $x$ act as particle positions, residual layers as external forces inducing velocities $v$; particles collide in training but simply cross in inference. It is worth noting that the only trainable component is $\theta$, which governs both the residual layer and the associated external force field.

For a layer with input $x \in \mathbb{R}^D$ and output $v \in \mathbb{R}^D$, instead of regarding it as one particle in $D$-dim space leads to no collision or $D$ particles in one-dim space collisions without stochastic direction changes, we introduce a hyper-parameter collision_heads, reshaping $x, v$ to collision_heads $\times N$ matrices $X, V$, and there are $N = \frac{D}{\text{collision\_heads}}$ particles colliding in collision_heads-dimensional space. $x_i, v_i \in \mathbb{R}^{\text{collision\_heads}}$, the $i$-th row of $X, V$, are initial position and velocity of particle $i$.

Similar to multi-head attention in Transformers that uses multiple heads to capture different aspects of feature relationships, our collision_heads mechanism controls the dimensional structure of the collision space where particles interact. However, unlike Transformers which partition the feature space into multiple independent representation subspaces, our approach consolidates all particles within one single collision_heads-dimensional space where inter-particle collisions occur collectively.

Specifically, our KITINet simulates the collisions by imitating the DSMC method:

1. Calculate the relative distance, the relative velocity, the center-of-mass position, and the center-of-mass velocity between $N$ particles:

$$
(X_r)_{i,j} = |\boldsymbol{x}_i - \boldsymbol{x}_j|, \quad (V_r)_{i,j} = |\boldsymbol{v}_i - \boldsymbol{v}_j|
$$
$$
(\boldsymbol{X}_{cm})_{i,j} = \frac{1}{2}(\boldsymbol{x}_i + \boldsymbol{x}_j), \quad (\boldsymbol{V}_{cm})_{i,j} = \frac{1}{2}(\boldsymbol{v}_i + \boldsymbol{v}_j). \tag{2}
$$

Note that each element in $X_r$ and $V_r$ is a scalar, while in $\boldsymbol{X}_{cm}$ and $\boldsymbol{V}_{cm}$ is a vector.

2. Simulate the change of velocity $\Delta\boldsymbol{V}$:

$$
(\Delta\boldsymbol{V})_{i,j} = (\boldsymbol{V}_{cm})_{i,j} + \frac{1}{2}(V_r)_{i,j}\boldsymbol{n}_{i,j} - \boldsymbol{v}_i, \quad (\Delta\boldsymbol{V})_{j,i} = (\boldsymbol{V}_{cm})_{j,i} + \frac{1}{2}(V_r)_{j,i}\boldsymbol{n}_{j,i} - \boldsymbol{v}_j, \tag{3}
$$

where $\boldsymbol{n}_{i,j}$ is a random vector distributed uniformly on the collision_heads-dim unit sphere, and $\boldsymbol{n}_{j,i} = -\boldsymbol{n}_{i,j}$. This expression builds on Equation (16). $(V_r)_{i,j}\boldsymbol{n}_{i,j}$ and $(V_r)_{j,i}\boldsymbol{n}_{j,i}$ are adapted from Equation (15) and are employed to compute the relative receding velocity after collision in the center-of-mass system.

3. We introduce a hyper-parameter coll_coef. For each pair $i, j$, accept the collision if

$$
\frac{(V_r)_{i,j} \cdot (U_r)_{i,j}}{v_r^{max}} > 1 - \text{coll\_coef}, \tag{4}
$$

where $(U_r)_{i,j} = e^{-(X_r)_{i,j}}$, $v_r^{max} = \max(V_r)$. This equation is based on Equation (14). Unlike the DSMC method, which divides space into cells and only permits collisions inside the cells, our approach permits collisions between any pair of particles. We introduce $U_r$, interpreted as the collision probability distribution under a unit mean free path. As $(X_r)_{i,j}$ increases, $(U_r)_{i,j}$ decreases, reducing the probability of the collision between pair $i, j$; conversely, as $(X_r)_{i,j}$ decreases, $(U_r)_{i,j}$ increases, making the collision more likely.

4. Update the velocity and position of the particles by the collision model $\boldsymbol{x}_i' = \boldsymbol{x}_i^* + dt * \boldsymbol{v}_i'$ where:

$$
\boldsymbol{v}_i' = \boldsymbol{v}_i + \sum_{j \text{ in accepted pair } i,j} (\Delta\boldsymbol{V})_{i,j}, \quad \boldsymbol{x}_i^* = \frac{1}{1+k}\left(\boldsymbol{x}_i + \sum_{j \text{ in accepted pair } i,j} (\boldsymbol{X}_{cm})_{i,j}\right), \tag{5}
$$

where $k$ is the number of accepted collisions of the $i$-th particle. $(\boldsymbol{X}_{cm})_{i,j}$ is the approximate collision position of pair $i, j$. $\boldsymbol{x}_i^*$ is the average of all collision positions of the $i$-th particle and its initial position. It is used to simulate the position change of particles $i$ during $dt$ time, which is negligible in DSMC. The necessity for position update and $\boldsymbol{x}_i^*$ will be discussed in Section 5.6.

The algorithm is summarized in Algorithm 1, as designed to be efficient and can be easily integrated into existing deep learning frameworks. Meanwhile, this algorithm still satisfies the assumptions of: homogeneous gas, particle symmetry, each pair of particles has an average collision probability of $\frac{2}{N(N-1)}$, molecular chaos and elastic collisions with no loss of energy or momentum. According to [18; 33; 2], it can still well approximate the behavior of the BTE process under these conditions.

The time complexity is $O(N^2 \cdot \text{collision\_heads}) = O(\frac{D^2}{\text{collision\_heads}})$. As $D$, the size of the feature vector, is a fixed parameter, introducing collision_heads may reduce the complexity of the KITINet module.

## 4 MECHNISTIC INSIGHTS OF KITINET COLLISION

A natural question arise: *What benefits does the introduction of KITINet collision bring to neural networks?* To address this question, we first demonstrate in Section 4.1 and Section 4.2 that KITINet collision induces parameter condensation, a phenomenon recognized as an indicator of strong model generalization capability [47; 49; 46]. Our analysis draws on the perspective of entropy in physics (Section 4.1) and a theoretical examination of a simplified case (Section 4.2). Furthermore, synthetic and real-world experiments in Section 5.7 provide empirical validation for this phenomenon. Finally, in Section 4.3, we clarify the distinction between KITINet and the dropout technique [41].

## 4.1 Physics analysis

In statistical mechanics and fluid dynamics, the viscosity coefficient $\eta$ is computed using the Green-Kubo relations [13; 20]

$$\eta = \frac{1}{VkT} \int_0^\infty \langle J(t)J(0) \rangle .$$ (6)

Just as macroscopic temperature is related to the microscopic kinetic energy of particles, Equation (6) establishes a bridge between the macroscopic viscosity coefficient $\eta$ and the microscopic particle stress flux $J$. When the interactions between particles consist solely of hard-sphere collisions, $J$ can be expressed as [44]:

$$J(t) = m \sum_{i=1}^N v_i^2(t) + \frac{1}{2} \sum_{i=1}^N \sum_{j=1}^N |\boldsymbol{F}_{coll,ij} \times (\boldsymbol{x_i} - \boldsymbol{x_j})|,$$ (7)

where $\boldsymbol{x_i}$ and $\boldsymbol{v_i}$ denote the position and velocity of the $i$-th particle, and $\boldsymbol{F}_{coll,ij}$ represents the force exerted on particle $i$ during a hard-sphere collision with particle $j$. Substituting Equation (7) into Equation (6) yields $\eta = \eta^K + \eta^{K \times C} + \eta^C$. As reported in [12], in DSMC the cross term $\eta^{K \times C} = 0$:

$$\eta = \eta^K + \eta^C,$$ (8)

where the kinetic contribution $\eta^K$ is precisely the Chapman-Enskog viscosity and $\eta^C$ corresponds to the collision-induced correction.

Thus, collisions effectively increase the macroscopic viscosity coefficient of the underlying particle system, thereby supplying an additional viscous force that promotes entropy production. By delegating part of the redistribution and relaxation dynamics to particle collisions, the network is relieved from the necessity of generating highly fluctuating external forces. Consequently, the external force field becomes smoother and exhibits reduced variability. Such smoothness in the external force field implies that the underlying input–output mapping can be captured without resorting to abrupt or irregular parameter adjustments, thereby yielding a more condense parameterization of the network.

## 4.2 On the theory of condensation

In Section J, we provide detailed theoretical analysis under the simplified condition of a two-layer overparameterized linear network solving a regression problem, in order to compare the performance with and without KITINet. To make the analysis on KITINet tractable, we assume a thermal equilibrium, an ideal physical state with constant temperature and retain the following Theorem 4.1.

**Theorem 4.1.** *Under the setting in Section J.1 and assume KITINet is under thermal equilibrium, the introduction of KITINet collisions changes the rapid convergence process to a two-phase process: (1) The norm of neuron first decays to a small scale, inducing rapid reorientation in the low weight regime. (2) The model converges to a sparse solution through a condensation-like dynamics.*

**Proof Sketch of Theorem 4.1.** In the circumstance without KITINet collision, Theorem J.1 proves that the model converge in exponential rate. In contrast, in the circumstance with KITINet collision, we first show that the iterative process in this simplified scenario is a Markov process. Subsequently, Theorem J.5 proves that at each step, the model's weights gradually decay in expectation until they eventually converge to a sparse solution, which completes the proof.

**Insights from Theorem 4.1.** Our results show that under our simplified condition, the introduction of KITINet collisions changes the convergence process, transitioning from direct rapid convergence to a distinct two-phase process. Compared to the network without collision that rapidly converges to a complex solution, our KITINet converges to a sparser solution in this simplified circumstance, thereby leading to improved generalization.

## 4.3 Comparison between KITINet and Dropout

In physics, collisions between particles cause their spatial distribution to become more dispersed. When applied to neural networks, a similar "collision mechanism" in the feature layer induces sparsity across feature dimensions, functioning similar to dropout regularization. Notably, unlike dropout which discards information directly, KITINet preserves all information while promoting sparsity by incorporating stochastic collision dynamics. For CV or NLP tasks where the precision requirements are relatively low, dropout can achieve satisfactory results. However, for PDE tasks that demand high precision, employing dropout would amplify computational errors. In contrast, KITINet effectively preserves computational information, which can help minimize the numerical errors. The performance comparison of the PDE -solving tasks is presented in Section 5.4.

## 5 EXPERIMENTS

### 5.1 LARGE LANGUAGE MODEL (LLM) PRE-TRAINING FROM SCRATCH

**Dataset & models**. We pre-train the GPT-2 series from 0.1B to 1.5B parameter models with the standard next token prediction loss. We replace the residual connection after the attention module with a KITINet layer, obtaining a new series of models named KITINet-GPT-2. Our training corpus is a 30B token high-quality composition of web text (FineWeb-edu lozhkov2024fineweb-edu), mathematics (MegaMath zhou2025megamath), and code (OpenCoder Huang2024OpenCoderTO), which reflects current state-of-the-art data curation practices. The evaluation of GPT-2 and KITINet-GPT-2 is based on a diverse set of challenging downstream benchmarks, including knowledge-intensive tasks (MMLU hendryckstest2021, ARC allenaiarc) and commonsense reasoning tasks (HellaSwag zellers2019hellaswag, WinoGrande ai2winogrande).

**Results**. The Table 1 summarizes the accuracy on testing benchmarks with standard error. The results demonstrate that KITINet provides consistent performance gains over the baseline model in the majority of evaluation scenarios. A key finding from our analysis of the training dynamics is the enhanced training efficiency: KITINet consistently reaches target accuracy levels with approximately 20% fewer training steps than the baseline, which covers the overheads of its additional computation during training. This provides evidence for the practical advantages of our proposed KITINet architecture.

Table 1: Accuracy ↑ of vanilla and with KITINet-plugin GPT2 models.

| Model | MMLU | ARC_C | ARC_E | HellaSwag | WinoGrande |
|---|---|---|---|---|---|
| GPT2 [37] | 24.9 (0.35) | 21.8 (1.11) | 43.3 (1.01) | 38.4 (0.49) | 50.9 (1.41) |
| KITI-GPT2 | **25.1** (0.35) | **22.6** (1.11) | **43.5** (1.02) | **38.8** (0.48) | **51.2** (1.40) |
| GPT2-medium [37] | **27.0** (0.37) | 26.5 (1.12) | 52.3 (1.02) | 46.0 (0.49) | 53.4 (1.40) |
| KITI-GPT2-med | 26.2 (0.36) | **27.6** (1.13) | **52.5** (1.02) | **46.3** (0.50) | **53.8** (1.40) |
| GPT2-large [37] | 25.9 (0.35) | 28.5 (1.29) | 57.5 (1.02) | 46.7 (0.50) | 55.2 (1.39) |
| KITI-GPT2-large | **26.1** (0.36) | **28.6** (1.30) | **57.7** (1.01) | **47.2** (0.50) | **55.7** (1.40) |
| GPT2-xl [37] | 26.6 (0.36) | 31.8 (1.31) | 62.2 (0.99) | 50.6 (0.51) | 58.2 (1.38) |
| KITI-GPT2-xl | **27.2** (0.37) | **31.9** (1.32) | **62.9** (0.99) | **51.0** (0.51) | **58.5** (1.39) |

### 5.2 LLM CONTINUED PRE-TRAINING

We conducted additional experiments for LLM involving mathematical reasoning, i.e. continued pre-training of GPT-2 on the Open-Web-Math dataset paster2023openwebmath. Initialized with official OpenAI GPT-2 weights, we continued training with two configurations: GPT-2 with and without an additional KITINet layer. Both models were trained for 30 billion tokens, after which we assessed their performance on standard mathematical evaluation benchmarks, including GSM8K cobbe2021gsm8k, MATH hendrycksmath2021, MATH_QA amini-etal-2019-mathqa and OCW lewkowycz2022solving. The evaluation metric is perplexity of correct answers. The results are displayed in the Table 2, where lower values indicate better performance. Our results demonstrate that the KITINet version consistently achieves substantial perplexity improvements over the baseline across most evaluation scenarios, with gains approximately equivalent to those obtained by training for 20% additional tokens.

Table 2: Perplexity ↓ of vanilla and with KITINet GPT2.

| Model | GSM8K | MATH | MATHQA | OCW |
|---|---|---|---|---|
| GPT2 [37] | 17.34 | 12.35 | 31.24 | 7.34 |
| KITI-GPT2 | **17.29** | **12.18** | **30.39** | **7.21** |
| GPT2-medium [37] | **13.72** | 11.18 | 23.68 | 6.35 |
| KITI-GPT2-med | 13.82 | **11.02** | **23.05** | **6.21** |
| GPT2-large [37] | 11.59 | 10.27 | 19.56 | 5.53 |
| KITI-GPT2-large | **11.20** | **10.06** | **18.81** | **5.40** |
| GPT2-xl [37] | 10.59 | 9.78 | 17.48 | 5.23 |
| KITI-GPT2-xl | **10.42** | **9.63** | **16.99** | **5.10** |

### 5.3 IMAGE CLASSIFICATION ON CIFAR

**Dataset and models**. We conduct our evaluations on CIFAR with 50K training images and 10K test images. Our training setup follows [15], including models in different configurations, e.g., ResNet-34, ResNet-50, ResNet-101, and ResNet-152. To balance performance and cost, we selectively integrate our module in the last stage of the ResNet architecture.

**Settings**. The models are trained by SGD with batch size 128, momentum coefficient 0.9, and weight decay $5 \times 10^{-4}$. The learning rate is initialized to 1 for quadratic integration matrix in the implementation of Dit-ResNet [24] and 0.1 for all other parameters and decayed by a factor of ten at the 80th and 120th epochs, completing training after 160 epochs. We apply standard augmentation to the images in training: padding with 4 pixels on each side, followed by a random $32 \times 32$ crop,

and random horizontal flipping. For evaluation, we use the original $32 \times 32$ resolution without augmentation. Following [24], we incorporate quadratic neurons specifically into the same layer.

**Results.** Table 3 compares performance on CIFAR between KITINet and the vanilla ResNet model [15], as well as one biologically plausible adaptation by [24], which mimics the nonlinear dendritic computations observed in cortical neurons. All models are independently trained in identical settings for fairness, with reported metrics w.r.t. optimal validation performance. Our experiments show that KITINet achieves improvements on both CIFAR-10 and CIFAR-100 without introducing additional trainable parameters. KITI-ResNet-34 matches the accuracy of ResNet-

Table 3: Accuracy of KITINet and ResNet-based models.

| Model | CIFAR10 | CIFAR100 | MFLOPs |
|---|---|---|---|
| ResNet-34 [15] | 94.48 | 77.97 (0.12) | 73.5 |
| Dit-ResNet-34 [24] | 94.45 | 78.14 (0.07) | 73.5 |
| KITI-ResNet-34 | **95.04** | **78.67** (0.10) | 73.6 |
| ResNet-50 [15] | 94.75 | 78.27 (0.09) | 83.7 |
| Dit-ResNet-50 [24] | 94.53 | 78.61 (0.05) | 83.7 |
| KITI-ResNet-50 | **95.18** | **78.75** (0.04) | 85.9 |
| ResNet-101 [15] | 94.71 | 78.39 (0.08) | 159.2 |
| Dit-ResNet-101 [24] | 94.98 | 78.88 (0.05) | 159.2 |
| KITI-ResNet-101 | **95.01** | **79.09** (0.03) | 161.3 |
| ResNet-152 [15] | 94.67 | 78.41 (0.07) | 234.7 |
| Dit-ResNet-152 [24] | 95.21 | 78.84 (0.04) | 234.7 |
| KITI-ResNet-152 | **95.67** | **79.48** (0.03) | 236.8 |

152 on CIFAR-100 (78.67% vs. 78.41%), suggesting that it enables more efficient feature learning compared to simply increasing network depth (KITI-ResNet-34 introduces only a 0.18% increase in FLOPs compared to ResNet-34). Furthermore, it outperforms other biologically-inspired architectures on the test sets, indicating good generalization ability.

## 5.4 LEARNING NEURAL OPERATOR FOR PDE-SOLVING

**Dataset and models.** For PDE, we consider benchmark equation families with varying discretizations to assess resolution generalization. Our datasets are generated following the procedure in Section F. The Fourier Neural Operator (FNO) [23] and Operator Transformer (OFormer) [22] are selected as the neural solvers. For a more detailed description, refer to Section E.

**Settings.** FNOs are trained using Adam with an initial learning rate of $10^{-3}$, batch size 20 and a total training epoch 1K. OFormers are trained using Adam with an initial learning rate of $1 \times 10^{-3}$, batch size 16 and 50K epochs.

**Results.** Table 4 compares vanilla and KITINet on PDE. Across a diverse set of challenging PDE benchmarks and an airfoil flow simulation, KITINet enhances both FNO and OFormer. When integrated into FNO, it reduces the Burgers' equation error by approximately $23.50\%$ and the Navier-Stokes error by about $5.63\%$, while on the heat equation it yields a $27.52\%$ improvement. Similarly, OFormer augmented with KITINet achieves a $5.52\%$ decrease

Table 4: Performance comparison between vanilla and with KITINet models on PDE-solving tasks.

| Problem | Model | MSE $\downarrow$ |
|---|---|---|
| Burgers' Equ. | FNO [23] | 0.00217 |
| | KITI-FNO | **0.00166** |
| NS Equation | FNO [23] | 0.12023 |
| | KITI-FNO | **0.11346** |
| Heat Equation | FNO [23] | 0.07054 |
| | KITI-FNO | **0.05113** |
| Airfoil* | OFormer [22] | 16.39461 |
| | KITI-OFormer | **15.49034** |

* Airfoil problem uses Root MSE measurement.

in RMSE on the airfoil problem. Figure 5 shows vanilla FNO and FNO with KITINet applied.

## 5.5 TEXT CLASSIFICATION ON IMDB AND SNLI

**Dataset & models.** For text classification, we use on two benchmark datasets: 1) IMDb [30] for sentiment classification, testing the model's natural language understanding capability; 2) SNLI [6] for natural language inference, assessing its ability to reason over sentence pairs. Both datasets are widely adopted for evaluating model performance in NLP tasks. BERT [9] is a pre-trained language model based on the transformer architecture, achieving good performance on

Table 5: Accuracy comparisons.

| Model | IMDb | SNLI |
|---|---|---|
| Bert-cased [9] | 91.45% | 89.28% |
| KITI-Bert-cased | **92.96%** | **90.26%** |
| Bert-uncased [9] | 93.42% | 89.02% |
| KITI-Bert-uncased | **94.53%** | **90.56%** |

a wide range of NLP tasks. We adopt BERT as our baseline and enhance it by integrating our KITINet architecture into BERT's final transformer layer. The resulting hybrid model, termed KITI-BERT, demonstrates improved effectiveness over the original framework.

**Settings.** We do experiments with two pre-trained model variants, including bert-base-cased and bert-base-uncased. We set the tokenizer corresponding to the pre-trained model to process input

tokens. Both KITI-Bert and Bert are trained by Adam with a batch size of 32 with the same random seed. We set the learning rate to $2 \times 10^{-5}$ and the total fine-tuning training epochs to 40.

**Results.** In Table 5, KITI-Bert shows improvements on both IMDb and SNLI, i.e. 1.65% and 1.10% using the pre-trained parameters of bert-base-cased, and 1.18% and 1.73% using the pre-trained parameters of bert-base-uncased. With the same number of parameters, KITINet outperforms.

### 5.6 HYPER-PARAMETER ANALYSIS

In DSMC, the change in position $x$ in a single time step $dt$ is typically small and negligible. Consequently, $x$ is treated as fixed while the velocity $v$ is updated via collisions. Yet to ensure KITINet can be reduced to a ResNet-like architecture, we set $dt = 1$, making the change in position in a time step non-negligible. Thus, an explicit update to the position $x_i^*$ for particle $i$ is introduced.

For FNO, we do ablation for position updates, keeping all other settings identical. Table 6 consistently shows that position updates outperform the non-updating variant across all equations, highlighting the effectiveness of this position update mechanism.

We analyze the impact of two additional hyperparameters collision_heads and coll_coef. For hyperparameter collision_heads, we evaluate values ranging from 1 to $2^{10}$ on the FNO model for the Burgers' equation, while holding all other settings constant. Figure 2(a) shows that collision_heads exerts a substantial influence on performance: well-chosen values of collision_heads lead to marked improvements on both the training and test sets, while poorly chosen values degrade accuracy.

For hyper-parameter coll_coef, we evaluate values from 0.1 to 0.9 on the FNO model for the NS equation and Heat equation, while holding all other settings constant. Figure 2(b) and Figure 6 show that coll_coef has a notable impact, and the best choice of coll_coef may vary greatly over tasks.

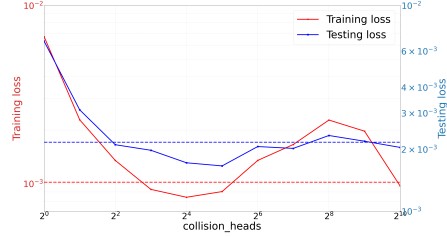

(a) Burgers' Equation

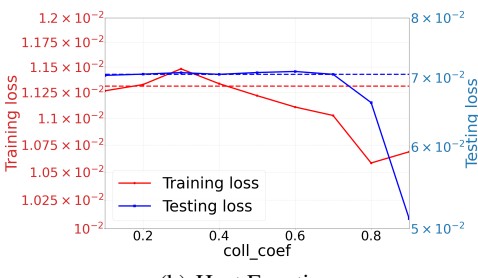

(b) Heat Equation

Figure 2: KITINet-FNO w/ different hyperparameters n_divide and coll_coef on Burgers' equation and Heat equation. The red and blue dashed lines show the performance of vanilla FNO as baselines.

### 5.7 FURTHER STUDY ON MODEL BEHAVIOR

In this section, we experimentally demonstrate the condensation phenomenon within KITINet, a phenomenon that is likely a key factor contributing to its superior performance and promising generalization ability. We first conduct analyses using a three-layer fully connected network and a six-layer skip-chain structured network. Furthermore, subsequent validation on both ResNet-18 and FNO consistently confirms this condensation effect.

Table 6: Comparing update and non-update mechanism FNO with KITINet on equations.

| Equation | Mechanism | MSE |
|----------|-----------|-----|
| Burgers' | non-update | 0.00173 |
|          | update | **0.00166** |
| NS | non-update | 0.11429 |
|    | update | **0.11346** |
| Heat | non-update | 0.05466 |
|      | update | **0.05113** |

**Synthesis experiments setup.** We consider the neural network with $d_{input}$ input and $d_{out}$ output dimensions. The dimension of the hidden neuron is set to the same value $m$. For both fully-connected and skip structures, they are initialized with all the parameters by a Gaussian $N(0, \sigma)$, where $\sigma = \frac{1}{m^\gamma}$. The size of the data is $n$. We construct the dataset from $\sum_{i=1}^{5} 3.5 \sin(5x_i + 1)$, where $\boldsymbol{x} = (x_1, x_2, x_3, x_4, x_5) \in \mathbb{R}^5$ and $x_i \in [-4, 2]$. $d_{input} = 5$ and $d_{output} = 1$. We fit the size of the training set $n = 80$ and $\gamma = 4$. This setting is used in [50] to analyze the condensation principle. For its generalizability, we use multiple activation functions i.e. ReLU, LeakyReLU, Sigmoid, and Tanh.

**Results on fully-connected network.** We employ a three-layer fully connected network with architecture $d_{input}$-m-$d_{output}$ as our baseline, where the second linear layer is replaced with our KITINet structure. As illustrated in Figure 3(a), KITINet significantly improves the condensation

extent of the model parameters. Other results are shown in Figure 7. Across all four common activation functions, KITINet consistently shows favorable behavior: maintaining robust parameter condensation or further enhancing the condensation effect compared to the original architecture. Furthermore, we compare the decrease in loss with different activation functions. On the training loss, KITINet performs slightly better (though the difference was minimal). On the test loss, KITINet achieves reductions of $6.8\%$, $7.5\%$, and $4.5\%$ respectively compared to the baseline when using ReLU, LeakyReLU, and Tanh activation functions, demonstrating its superior generalization capability.

**Results on skip-connection neural network.** Skip connections have become a core design in modern deep neural networks [15; 22; 9]. We design a six-layer baseline network where each layer incorporates skip connections. To systematically evaluate KITINet's effectiveness, we conduct comparative experiments by replacing: (1) only the last layer and (2) the last two layers with our KITINet structure. Our results suggest two key findings: First, KITINet consistently accelerates parameter condensation over conventional skip-connections. Second, replacing the last two layers with KITINet yields faster condensation by modifying only the final layer (see Figure 3(b)). This hierarchical improvement suggests that KITINet's benefits are cumulative when applied across multiple network layers. We also show more experimental results about evolution of the parameter condensation effect in Section H.

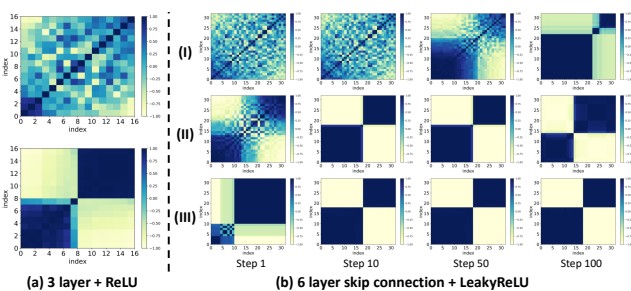

Figure 3: Results of parameter condensation across network configurations on synthetic data. (a) Top: Condensation patterns in 3-layer FC-ReLU networks; Bottom: Enhanced condensation after replacing the final layer with KITINet architecture. (b) Evolution of parameter condensation on a Six-layer skip-connected Network with LeakyReLU activation function. (Row I) without applying KITINet. (Row II) applying KITINet architecture on the last layer. (Row III) applying KITINet architecture on the last two layers. We choose the evolutionary trajectories at four critical checkpoints ($t \in \{1, 10, 50, 100\}$) to characterize the phase transitions and train 100 epochs. Our observation demonstrates that the KITINet structure facilitates faster and more effective parameter condensation.

**Results on Real-world Experiments.** For real-world tasks, KITINet also promotes consolidation. We employ the same experimental setting as described in Section 5 and utilize the average absolute cosine similarity to measure the degree of condensation

Table 7: Condensation degree $\rho$ of the Navier-Stokes Equation between Vanilla and KITINet.

| layer | conv0 | conv1 | conv2 | linear0 | linear1 | linear2 |
|---|---|---|---|---|---|---|
| FNO | **0.113** | 0.089 | 0.092 | 0.188 | 0.175 | 0.182 |
| KITI-FNO | 0.102 | **0.164** | **0.130** | **0.206** | **0.192** | **0.191** |

denoted by $\rho$ in a convolutional layer or a fully connected layer. This metric was adopted in [17] and the formal definition is provided in Section I. The results are promising. In the PDE task of solving naiver-stokes equation, as shown in Table 7, we observed that after adopting KITINet approach, the degree of condensation increased among most layers. In the image classification task, we observed the degree of condensation across different convolutional layers of ResNet18 and KITINet 18 trained on CIFAR-100. After applying KITINet, the first conv layer (conv1, transforming RGB to features) shows a notable improvement in condensation degree (from 0.235 to 0.248, improved by $5.3\%$), while the changes in condensation degree for the remaining layers are minimal. The results show that KITINet facilitates condensation, which may contribute to its enhanced generalization capability.

## 6 CONCLUSION AND LIMITATION FOR FUTURE WORK

We have introduced KITINet, leveraging the principles of kinetic theory to enhance the performance of neural networks. By simulating particle dynamics and incorporating collision-like interactions, KITINet is designed to achieve improved generalization capabilities and parameter condensation. Our experimental results demonstrate its effectiveness across various tasks. We also provide a mechanistic analysis to elucidate the underlying principles responsible for its superior performance. When ideal computing resources are available, future work will focus on further optimizing KITINet and exploring its applications in other domains, as well as more scaled benchmarks.

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

## A    RELATED WORK

**Residual Learning and Dynamical Systems.**  ResNet [15] introduced residual connections to mitigate vanishing gradients in deep networks. Subsequent studies reinterpreted residual networks through dynamical systems theory, with neural ODEs [7] modeling continuous-depth networks as ordinary differential equations (ODEs). ODE-RNN [38] simulates continuous dynamics of hidden states in RNNs. Neural controlled differential equations [19] extend this framework to incorporate control mechanisms, enabling adaptive feature propagation. Other extensions include second-order residuals [35] and flow models [3]. While these works provide valuable insights into the dynamics of residual learning, they primarily focus on deterministic, collision-free dynamics, neglecting the stochastic interactions and energy dissipation mechanisms inherent in real-world particle systems.

**Physics-Inspired Network Architectures**  Recent efforts integrate physical principles into neural architectures to enhance interpretability and data efficiency. Hamiltonian networks [14] preserve energy conservation laws, and Lagrangian networks [42] derive updates from variational principles. PDE-inspired models, such as PDE-GCN [11] and PDE-Net [26], parameterize spatial-temporal evolution via partial differential equations. Closest to our work, [45] proposed a convection-diffusion network (COIN), which incorporates diffusion layers after the ResNet architecture. But their formulation lacks explicit ties to residual learning or parameter condensation. Critically, while the above frameworks borrow mathematical structures from physics, they do not simulate collisional processes or exploit thermodynamic relaxation for network sparsity.

## B    MOLECULE DYNAMICS AND NEURAL ODE

Molecular dynamics (MD) [16] is a numerical method that simulates the motion of molecular or atomic systems at a microscopic level. Considering $N$ particles in one-dimensional space $1, .., N$, where the mass of the $i$-th particle is $m_i$, and its position is $x_i$. The potential energy is $V(x_1, ..., x_N)$. According to Newton's second law, we can write a second-order ordinary differential equation (ODE)

$$m_i \frac{d^2 x_i}{dt^2} = F_i = -\nabla_{x_i} V \tag{9}$$

where $F_i$ represents the net external force acting on particle $i$, including weak interaction forces (van der Waals forces), electromagnetic forces (Coulomb forces, chemical bonds), etc. Specifically, the expression for Neural ODE (NODE) [7] involves a first-order ODE:

$$\frac{d\boldsymbol{x}}{dt} = v(\boldsymbol{x}(t), t, \theta) \tag{10}$$

where $\boldsymbol{x} \in \mathbb{R}^D$ represents the hidden layer output; $t \in \mathbb{R}_+$ represents the network depth, while $v$ represents the neural network with parameters $\theta$. This can be analogized to $N$ particles moving with velocity $v$. For NODE, its corresponding Newtonian equation can be written as:

$$\frac{d^2 \boldsymbol{x}}{dt^2} = \frac{Dv}{Dt} = (\nabla_{\boldsymbol{x}} v)v + \frac{\partial v}{\partial t} := F/m \tag{11}$$

It can be regarded as a neural molecular dynamics system, where the hidden layer output $\boldsymbol{x}$ is particle position, and velocity $v$ is the derivative of the position parameterized by learnable parameters.

The NODE is a continuous model, implemented by discrete numerical methods, e.g., the Euler method, the Runge-Kutta method. The NODE can be trained by backpropagation through the adjoint method. It can be used to solve the problem of vanishing gradient and exploding gradient, and has wide applications in the field of machine learning, e.g., image generation, time series prediction, etc.

## C    DIRECT SIMULATION MONTE CARLO (DSMC)

In DSMC, we consider the hard sphere model, where the collision probability is proportional to the relative velocity of the particle pairs:

$$P_{\text{coll}}[i,j] = \frac{|\boldsymbol{v}_i - \boldsymbol{v}_j|}{\sum_{m=1}^{N_c} \sum_{n=1}^{m-1} |\boldsymbol{v}_m - \boldsymbol{v}_n|}, \tag{12}$$

where $N_c$ is the number of particles in the cell; the velocity $\boldsymbol{v}$ is proportional to the momentum $\boldsymbol{p}$ if assuming the particle mass is constant. The denominator is expensive to compute, so the DSMC method uses a rejection sampling method to approximate the collision probability:

1. Estimate the number of candidate collision pairs $M_{\mathrm{cand}}$ by the no-time-counter method [1]:

$$M_{\mathrm{cand}} = \frac{N_c(N_c - 1)F_N \pi d^2 v_r^{\max} \tau}{2V_c},$$  (13)

where $d$ is the particle diameter, the $v_r^{\max}$ is the estimated maximum relative velocity, the $\tau$ is the time step, the $V_c$ is the cell volume.

2. Random select $M_{\mathrm{cand}}$ pairs of particles. For each pair $i, j$, generate a random number $\Re_1$ from the uniform distribution $U(0, 1)$, and accept the collision if

$$|\boldsymbol{v}_i - \boldsymbol{v}_j|/v_r^{\max} > \Re_1.$$  (14)

3. If the collision is accepted, update the velocity of the particles according to the collision model, with position unchanged.

4. Repeat the above steps for all cells, then proceed to the next time step.

The hard sphere model is a hard-body collision. The particles conserve momentum and energy and scatter off in a random direction. Set post-collision relative velocity in a polar coordinate system:

$$\boldsymbol{v}_r^* = v_r[(\sin\theta\cos\phi)\hat{\boldsymbol{x}} + (\sin\theta\sin\phi)\hat{\boldsymbol{y}} + \cos\theta\,\hat{\boldsymbol{z}}],$$  (15)

and the angle are set as $\phi = 2\pi\Re_2$ and $\theta = \cos^{-1}(2\Re_3 - 1)$, where $\Re_2$ and $\Re_3$ are random numbers from the uniform distribution $U(0, 1)$. Denote the center of mass velocity as $\boldsymbol{v}_{\mathrm{cm}} = (\boldsymbol{v}_i + \boldsymbol{v}_j)/2$, then the post-collision velocity can be calculated as:

$$\boldsymbol{v}_i' = \boldsymbol{v}_{\mathrm{cm}} + \boldsymbol{v}_r^*/2, \boldsymbol{v}_j' = \boldsymbol{v}_{\mathrm{cm}} - \boldsymbol{v}_r^*/2,$$  (16)

## D  ALGORITHMS

Besides the KITINet architecture we have introduced, we also experimented with an alternative architecture of KITINet, a-edition KITINet. In physics, acceleration can exhibit abrupt changes due to external forces, whereas velocity should vary continuously. Therefore, a-edition KITINet considers residual connections as position $\boldsymbol{x}$ and residuals as acceleration $\boldsymbol{a}$, requiring velocity $\boldsymbol{v}$ from previous a-edition KITINet, and outputs $\boldsymbol{x}'$ and $\boldsymbol{v}'$. For the first layer of a-edition KITINet, $\boldsymbol{v}$ would be a random variable drawn from a Gaussian distribution, which satisfies the thermodynamic distribution. During the a-edition KITINet, the initial velocity for collision simulation would be $\boldsymbol{v} + dt * \boldsymbol{a}$, and the velocity after collision $\boldsymbol{v}'$ is recorded for the next a-edition KITINet.

From a physics perspective, the a-edition KITINet more faithfully satisfies Newton's second law with Equation (9) and **BTE** with Equation (1); from a neural-network perspective, the variable $\boldsymbol{v}$ within the network functions analogously to an RNN's hidden state, storing and propagating information. However, in experiments, the a-edition KITINet failed to deliver satisfactory results.

# E PDE-SOLVER ARCHITECTURE

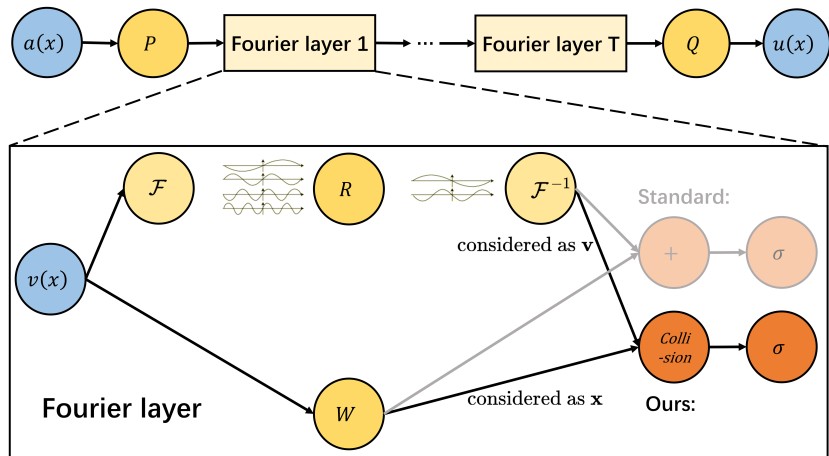

Figure 4: The full architecture of Fourier Neural Operators (FNO) with and without KITINet applied.

## E.1 FOURIER NEURAL OPERATOR (FNO)

FNO [23] is a neural operator that implements a resolution-invariant global convolution by FFT'ing input features, applying a learnable linear transform to a truncated set of frequency modes, and then inverse-FFT'ing back to the spatial domain. It efficiently captures long-range dependencies and generalizes across discretizations. As Fig. 4 in the appendix shows, for each Fourier layer with KITINet applied, the outputs of the Fourier convolution are considered as $v$, while the outputs of the linear transformation are considered as $x$.

## E.2 OPERATOR TRANSFORMER (OFORMER)

OFormer [22] embeds Fourier neural operator blocks into a Transformer-style sequence model, applying FFTs to input tokens, learnable complex-valued multipliers on truncated frequency modes, and inverse FFTs back to space, while its attention mechanism enables these spectral operations to be conditioned on arbitrary, irregular input locations, making it directly applicable to non-uniform and unstructured grids. In the Transformer architecture with KITINet applied, the outputs of the self-attention and MLP layers are considered as $v$, while the residual connections are considered as $x$.

# F PDE DATSETS GENERATION

## F.1 BURGERS' EQUATION

The one-dimensional Burgers' equation is a nonlinear PDE commonly used to describe viscous fluid flow in a single spatial dimension. It takes the form (We use dataset from [23]):

$$\partial_t u(x,t) + \partial_x(u^2(x,t)/2) = \nu \partial_{xx} u(x,t), \quad x \in (0,1), t \in (0,1],$$

$$u(x,0) = u_0(x), \quad x \in (0,1).$$

The initial condition $u_0(x)$ is generated according to $u_0 \frown \mu$ where $\mu = \mathcal{N}(0, 625(-\Delta + 25I)^{-2})$ with periodic boundary conditions and the viscosity is set to $\nu = 0.1$. Fourier Neural Operators are chosen for solving this equation, learning the operator mapping the initial condition to the solution at time one, $G^\dagger : L^2_{per}((0,1); \mathbb{R}) \to H^r_{per}((0,1); \mathbb{R})$ defined by $u_0 \mapsto u(\cdot, 1)$ for any $r > 0$.

## F.2 NAVIER-STOKES (NS) EQUATION

The two-dimensional NS equation for a viscous, incompressible fluid in vorticity form on the unit torus takes the form:

$$\partial_t \omega(x,t) + u(x,t) \cdot \nabla \omega(x,t) = \nu \Delta \omega(x,t) + f(x), \quad x \in (0,1)^2, t \in (0,T],$$

$$\nabla \cdot u(x,t) = 0, \quad x \in (0,1)^2, t \in (0,T],$$

$$\omega(x,0) = \omega_0(x), \quad x \in (0,1)^2.$$

The initial condition $\omega_0(x)$ is generated according to $\omega_0 \frown \mu$ where $\mu = \mathcal{N}(0, 7^{3/2}(-\Delta + 49I)^{-2.5})$ with periodic boundary conditions, the force $f(x) = 0.1(\sin(2\pi(x_1 + x_2)) + \cos(2\pi(x_1 + x_2)))$

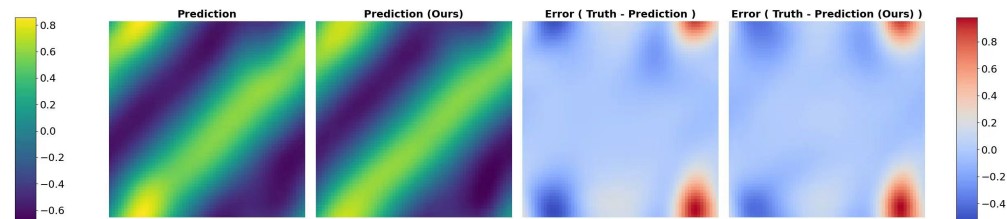

Figure 5: FNOs' performance on NS equation, both vanilla and with KITINet applied. Left two: FNOs' predictions at the final time step; Right two: their corresponding absolute error maps.

and the viscosity is set to $\nu = 1e{-}3$. FNOs are chosen for this equation, learning the operator mapping the vorticity up to time 10 to the solution up to $T > 10$, $G^{\dagger} : C([0, 10]; H_{per}^r((0, 1); \mathbb{R})) \rightarrow C([10, T]; H_{per}^r((0, 1)); \mathbb{R})$ defined by $\omega|_{(0,1)^2 \times (0,10]} \mapsto \omega|_{(0,1)^2 \times (10,T]}$ for any $r > 0$. All data are generated on a $256 \times 256$ grid with a pseudospectral method and are downsampled to $32 \times 32$ or $64 \times 64$. The resolution is fixed to $32 \times 32$ for training and $64 \times 64$ for testing.

### F.3 HEAT EQUATION

The two-dimensional Heat equation for a heated square box form on the unit torus takes the form:

$$\partial_t u(x, t) = \alpha \Delta u(x, t) + q(x), \quad x \in (0, 1)^2, t \in (0, T],$$

$$u(x, 0) = u_0(x), \quad x \in (0, 1)^2.$$

The initial condition $u_0(x)$ is generated according to $u_0 \backsim \mu$ where $\mu = \mathcal{N}(0, 7^{3/2}(-\Delta + 49I)^{-2.5})$ with periodic boundary conditions, the heat source $q|_{\partial\Omega} = 0.1$ and the thermal diffusivity is set to $\alpha = 1e{-}4$. Here Fourier Neural Operators are chosen for solving this equation, learning the operator mapping the vorticity up to time 10 to the solution up to some later time $T > 10$, $G^{\dagger} : C([0, 10]; H_{per}^r((0, 1); \mathbb{R})) \rightarrow C([10, T]; H_{per}^r((0, 1)); \mathbb{R})$ defined by $\omega|_{(0,1)^2 \times (0,10]} \mapsto \omega|_{(0,1)^2 \times (10,T]}$ for any $r > 0$. All data are generated on a $256 \times 256$ grid with a pseudospectral method and are downsampled to $32 \times 32$ or $64 \times 64$. The resolution is fixed to be $32 \times 32$ for training and $64 \times 64$ for testing.

### F.4 AIRFOIL

For this problem, we study the two-dimensional time-dependent compressible flow around the cross-section of airfoils, with different inflow speeds (Mach numbers) and angles of attack, and NS equation is also used to describe the problem. Here Operators Transformer are chosen for this problem, learning the mapping the velocity up to time $0.576s$ to the solution up to $T = 4.8s$, $G^{\dagger} : \boldsymbol{u}(\cdot, t)|_{t \in [0, 0.576]} \mapsto \boldsymbol{u}(\cdot, t)|_{t \in (0.576, 4.800]}$. All data on irregular grids are generated by [36], with conventional solver SU2 [10].

# G    ADDITIONAL EXPERIMENT ABOUT HYPER-PARAMETER ANALYSIS

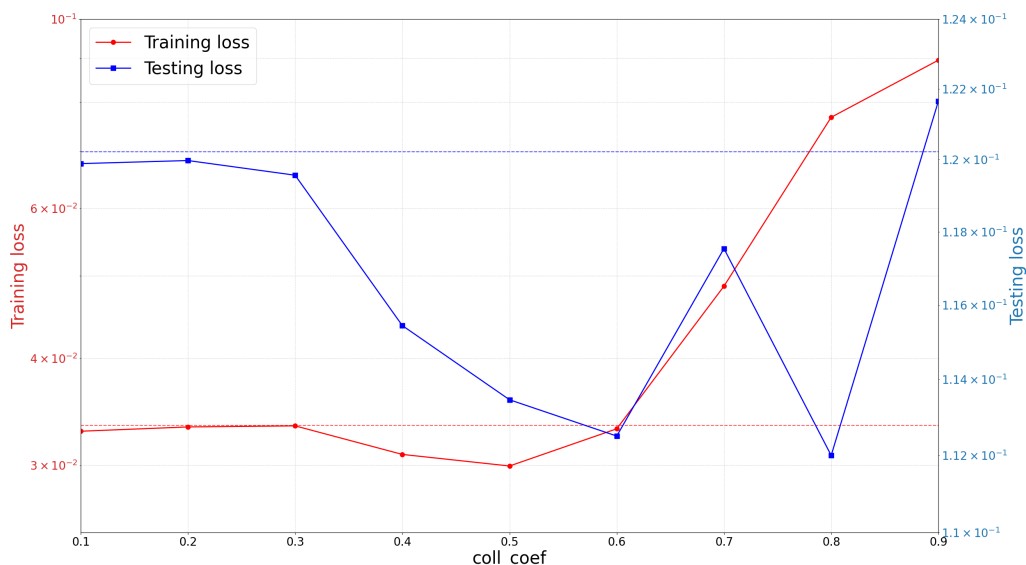

Figure 6: The performance of KITI-FNO with different hyper-parameter coll_coef on NS equation. The red and blue dashed lines show the performance of vanilla FNO as baselines.

# H    ADDITIONAL EXPERIMENT ABOUT CONDENSATION

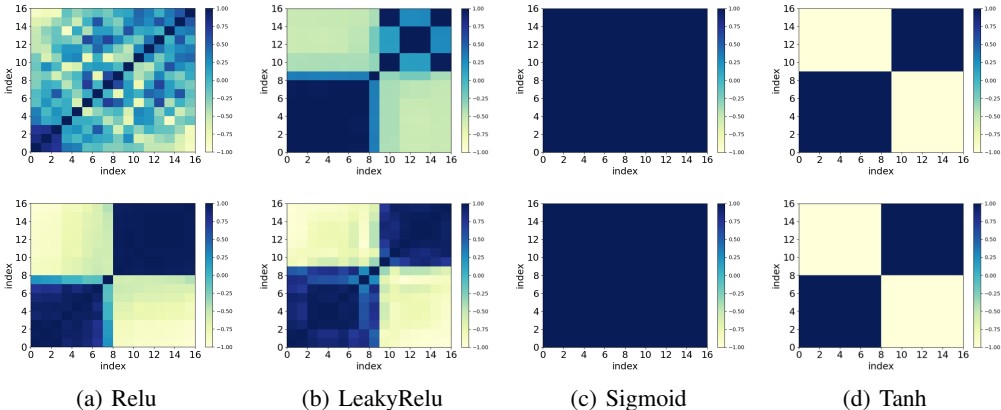

(a) Relu          (b) LeakyRelu          (c) Sigmoid          (d) Tanh

Figure 7: Results of parameter condensation on Three-layer Fully-connected Network. (Row 1) linear networks versus (Row 2) KITINet -incorporated networks. Systematic validation is performed across four activation functions: ReLU, LeakyReLU, Sigmoid, and Tanh.

# I    THE FORMAL DEFINITION OF THE DEGREE OF CONDENSATION.

In this section, we provide the formal definitions of condensation for fully connected networks and convolutional networks, which are given in Definition 3.1 and 3.2 .

**Definition I.1** (Weight Correlation in FCN). Given weight matrix $w_{\mathfrak{l}} \in \mathbb{R}^{N_{\mathfrak{l}-1} \times N_{\mathfrak{l}}}$ of the $\mathfrak{l}$-th layer, the average weight correlation is defined as

$$\rho(w_{\mathfrak{l}}) = \frac{1}{N_{\mathfrak{l}}(N_{\mathfrak{l}} - 1)} \sum_{\substack{i,j=1 \\ i \neq j}}^{N_{\mathfrak{l}}} \frac{|w_{\mathfrak{l}i}^T w_{\mathfrak{l}j}|}{||w_{\mathfrak{l}i}||_2 ||w_{\mathfrak{l}j}||_2}, \tag{17}$$

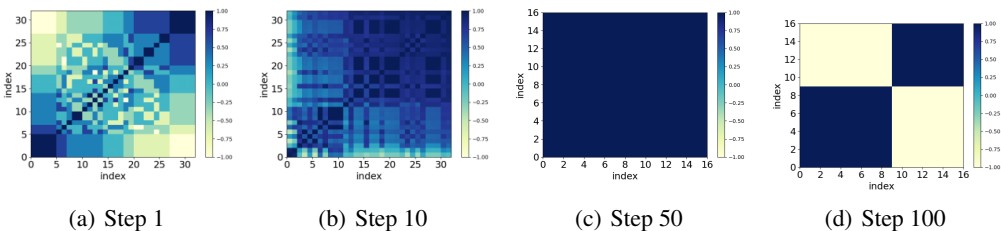

(a) Step 1      (b) Step 10      (c) Step 50      (d) Step 100

Figure 8: Evolution of parameter condensation effect on Six-layer ReLU skip-connected network without applying KITINet architecture. The process of paramter condensation is relatively slow.

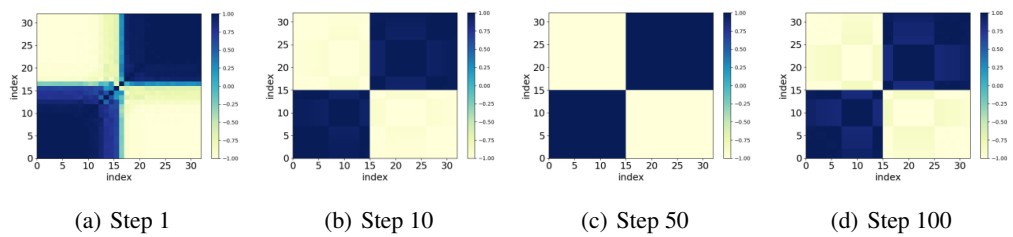

(a) Step 1      (b) Step 10      (c) Step 50      (d) Step 100

Figure 9: Evolution of parameter condensation effect on Six-layer ReLU skip-connected network applying KITINet architecture on the last layer. The process of parameter condensation is relatively faster.

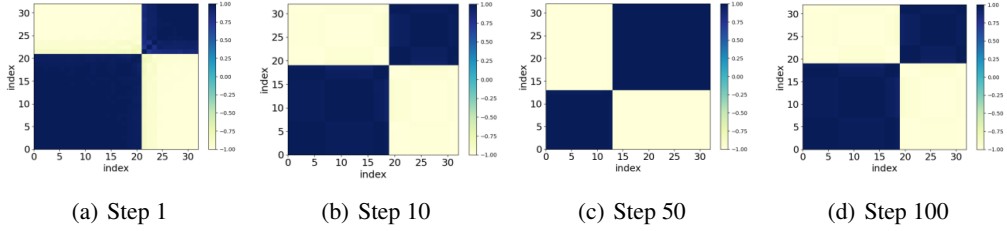

(a) Step 1      (b) Step 10      (c) Step 50      (d) Step 100

Figure 10: Evolution of parameter condensation on Six-layer skip-connected Network applying KITINet architecture on the last two layers. The process of parameter condensation is significantly much faster and stable.

where $w_{\mathfrak{l}i}$ and $w_{\mathfrak{l}j}$ are $i$-th and $j$-th column of the matrix $w_{\mathfrak{l}}$, corresponding to the $i$-th and $j$-th neuron at $\mathfrak{l}$-th layer, respectively. Intuitively, $\rho(w_{\mathfrak{l}})$ is the average cosine similarity between weight vectors of any two neurons at the $\mathfrak{l}$-th layer.

**Definition I.2** (Weight Correlation in CNN). Given the filter tensor $\mathbf{w}_{\mathfrak{l}} \in \mathbb{R}^{f \times f \times N_{\mathfrak{l}-1} \times N_{\mathfrak{l}}}$ of the $\mathfrak{l}$-th layer, where $f \times f$ is the size of the convolution kernel, $\mathbf{w}_{\mathfrak{l}i} \in \mathbb{R}^{f \times f \times N_{\mathfrak{l}-1}}$ and $\mathbf{w}_{\mathfrak{l}j} \in \mathbb{R}^{f \times f \times N_{\mathfrak{l}-1}}$ are the $i$-th and $j$-th filter, respectively, of the filter tensor $\mathbf{w}_{\mathfrak{l}}$. By reshaping $\mathbf{w}_{\mathfrak{l}i}$ and $\mathbf{w}_{\mathfrak{l}j}$ into $\mathbf{w}'_{\mathfrak{l}i} \in \mathbb{R}^{f^2 \times N_{\mathfrak{l}-1}}$ and $\mathbf{w}'_{\mathfrak{l}j} \in \mathbb{R}^{f^2 \times N_{\mathfrak{l}-1}}$, respectively, the weight correlation is defined as

$$\rho(\mathbf{w}_{\mathfrak{l}}) = \frac{1}{N_{\mathfrak{l}}(N_{\mathfrak{l}}-1)N_{l-1}} \sum_{\substack{i,j=1 \\ i \neq j}}^{N_{\mathfrak{l}}} \sum_{z=1}^{N_{l-1}} \frac{|\mathbf{w}'^{T}_{\mathfrak{l}i,z} \mathbf{w}'_{\mathfrak{l}j,z}|}{||\mathbf{w}'_{\mathfrak{l}i,z}||_2 ||\mathbf{w}'_{\mathfrak{l}j,z}||_2}, \tag{18}$$

where $\mathbf{w}'_{\mathfrak{l}i,z}$ and $\mathbf{w}'_{\mathfrak{l}j,z}$ are the $z$-th column of $\mathbf{w}'_{\mathfrak{l}i}$ and $\mathbf{w}'_{\mathfrak{l}j}$ respectively. Intuitively, $\rho(\mathbf{w}_{\mathfrak{l}})$ is defined as the cosine similarity between filter matrices.

# J A THEORY ANALYSIS OF CONDENSATION.

## J.1 SETTING.

First, we present a formulation of the problem statement and establish the necessary notation that will be used throughout our theoretical analysis. Consider a linear regression problem $y_i = \boldsymbol{k}\boldsymbol{x_i}, i = \{1, 2, \cdots, n\}$, where the training data $\boldsymbol{x_i} \in \mathbb{R}^d \sim N(0, I)$, $\mathbf{k} \in \mathbb{R}^{1 \times d}$. We use $y = \boldsymbol{a}^\top \boldsymbol{W} \boldsymbol{x}$ to retain the $\boldsymbol{k}$. We set $\boldsymbol{W}$ is a $m \times d$ matrix and $\boldsymbol{a}$ is a $1 \times m$ matrix. We use $\boldsymbol{W}(t), \boldsymbol{a}(t)$ to denote the value of $\boldsymbol{W}, \boldsymbol{a}$ at step $t$.

Let $\boldsymbol{W} = \begin{bmatrix} \boldsymbol{w}_1^\top \\ \boldsymbol{w}_2^\top \\ \vdots \\ \boldsymbol{w}_m^\top \end{bmatrix}$, $\boldsymbol{w}_i \in \mathbb{R}^{d \times 1}$, we assume that $w_{i,j}(t) \sim N(0, \sigma_1(t))$ at step $t$ for every $i \in [m], j \in [d]$. Let $\boldsymbol{a} = \begin{bmatrix} a_1 & a_2 & \cdots & a_m \end{bmatrix}$. we assume that $a_i(t) \sim N(0, \sigma_2(t))$ at step $t$ for every $i \in [m]$.

We consider MSE-loss as follow:

$$\min_{\boldsymbol{W}, \boldsymbol{a}} L = \frac{1}{2n} \sum_{i=1}^{n} (\boldsymbol{a}^\top \boldsymbol{W} \boldsymbol{x_i} - y_i)^2 \tag{19}$$

Let $\boldsymbol{\theta}(t) = \boldsymbol{a}(t)^\top \boldsymbol{W}(t)$ and $\boldsymbol{\theta}^* = \boldsymbol{k}$. We assume that the nueral network is over-parametrized, i.e. $m \gg d$. Let $\eta$ denote the learning rate.

## J.2 CASE 1: THE CIRCUMSTANCE WITHOUT COLLISION.

**Theorem J.1.** *Under the setting in Section J.1, the convergence rate of the model parameters is exponential.*

**Proof of Theorem J.1.**

The gradient is

$$\frac{\partial L}{\partial \boldsymbol{w}_l^\top} = a_l^\top \cdot \frac{1}{n} \sum_{i=1}^{n} (\boldsymbol{a}^\top \boldsymbol{W} \boldsymbol{x_i} - y_i) \cdot \boldsymbol{x}_i^\top \tag{20}$$

$$\frac{\partial L}{\partial a_l^\top} = \boldsymbol{w}_l^\top \cdot \frac{1}{n} \sum_{i=1}^{n} (\boldsymbol{a}^\top \boldsymbol{W} \boldsymbol{x_i} - y_i) \cdot \boldsymbol{x}_i^\top \tag{21}$$

By using GD, we have

$$\boldsymbol{W}(t+1) = \boldsymbol{W}(t) - \eta \cdot \boldsymbol{a}(t)^\top \cdot \frac{1}{n} \sum_{i=1}^{n} (\boldsymbol{a}^\top(t) \boldsymbol{W}(t) \boldsymbol{x_i} - y_i) \cdot \boldsymbol{x}_i^\top \tag{22}$$

$$\boldsymbol{a}(t+1) = \boldsymbol{a}(t) - \eta \cdot \boldsymbol{W}(t)^\top \cdot \frac{1}{n} \sum_{i=1}^{n} (\boldsymbol{a}(t)^\top \boldsymbol{W}(t) \boldsymbol{x_i} - y_i) \cdot \boldsymbol{x}_i^\top \tag{23}$$

By multiplying the above two equations we get

$$\boldsymbol{\theta}(t+1) = \boldsymbol{\theta}(t) - \eta \cdot \boldsymbol{a}(t) \cdot \boldsymbol{a}(t)^\top \cdot \frac{1}{n} \sum_{i=1}^{n} (\boldsymbol{a}(t)^\top \boldsymbol{W}(t) \boldsymbol{x_i} - y_i) \cdot \boldsymbol{x}_i^\top$$

$$- \eta \cdot \frac{1}{n} \sum_{i=1}^{n} (\boldsymbol{a}(t)^\top \boldsymbol{W}(t) \boldsymbol{x_i} - y_i) \cdot \boldsymbol{x}_i^\top \cdot \boldsymbol{W}(t)^\top \boldsymbol{W}(t) + O(\eta^2)$$

By Large Number Law, we have

$$\boldsymbol{a}(t)\boldsymbol{a}(t)^\top = \sum_{i=1}^{m} a_i(t)^2 \approx \sigma_2(t) \cdot m \tag{24}$$

$$\boldsymbol{W}(t)^\top \boldsymbol{W}(t) = \sum_{i=1}^{m} \boldsymbol{w}_i(t)\boldsymbol{w}_i(t)^\top \approx \sum_{i=1}^{m} \sigma_1(t) \cdot I = m \cdot \sigma_1(t) \cdot I \tag{25}$$

Since $y_i = \boldsymbol{\theta}^* \cdot \boldsymbol{x}_i$, we can approximately get

$$\boldsymbol{\theta}(t+1) - \boldsymbol{\theta}^* \approx \boldsymbol{\theta}(t) - \boldsymbol{\theta}^* - \eta \cdot (1 + m \cdot (\sigma_1(t) + \sigma_2(t))) \cdot (\boldsymbol{\theta}(t) - \boldsymbol{\theta}^*) \cdot \frac{1}{n}\sum_{i=1}^{n} \boldsymbol{x}_i \cdot \boldsymbol{x}_i^\top \tag{26}$$

By large number theorem, when n is sufficiently large,

$$\frac{1}{n}\sum_{i=1}^{n} \boldsymbol{x}_i \cdot \boldsymbol{x}_i^\top \approx E[\boldsymbol{x}_i \cdot \boldsymbol{x}_i^\top] = I \tag{27}$$

So finally we retain

$$\boldsymbol{\theta}(t+1) - \boldsymbol{\theta}^* \approx (1 - \eta \cdot (1 + m \cdot (\sigma\_1(t) + \sigma\_2(t))))(\boldsymbol{\theta}(t) - \boldsymbol{\theta}^*) \tag{28}$$

$$\approx (\boldsymbol{\theta}(0) - \boldsymbol{\theta}^*) \cdot \prod_{i=0}^{t} \beta(t) \tag{29}$$

where $\beta(t) = 1 - \eta \cdot (1 + m \cdot (\sigma_1(t) + \sigma_2(t)))$.

Therefore, under collision-free conditions, the model can rapidly descend to convergence in exponential time as long as $\eta$ is small and properly selected. $\qquad\square$

### J.3   CASE 2: THE CIRCUMSTANCE WITH COLLISION (KITINET).

Our key assumption here is that KITINet is under thermal equilibrium. In this idealized state, the temperature is uniform and constant within the system, and all particles have the same velocity distribution and same probability to collide. Here, we inject uniform constant noise into $a$ as the simulation for thermal equilibrium. While this simplification differs from the real collision dynamics, it can provide valuable insights into how KITINet enhances model robustness.

**Theorem J.2.** *Under the setting in Section J.1 and assume KITINet is under thermal equilibrium, the introduction of KITINet collisions changes the convergence process to a two-phase process:*
*(1) The norm of neuron in $W_1$ first decays to a small scale, inducing rapid reorientation in the low weight regime.*
*(2) The model converges to a sparse solution through a condensation-like dynamics.*

**Proof of Theorem J.2.**

The second phase that the convergence behavior under small-weight regimes has been extensively studied [31; 28; 49; 32; 29], so we only provide the proof in the first phase.

The update rule to simulate the thermal equilibrium KITINet collision is modeled as follow: for every neuron $\boldsymbol{w}_i$ and $a_i$ at step $t$, we have

$$\boldsymbol{w}_i(t+1) = \boldsymbol{w}_i(t) - \eta \cdot a_i(t) \cdot \frac{1}{n}\sum_{i=1}^{n}(\boldsymbol{a}(t)^\top \boldsymbol{W}(t)\boldsymbol{x}_i - y_i) \cdot \boldsymbol{x}_i \tag{30}$$

$$a_i(t+1) = a_i(t) + \delta_i(t) \tag{31}$$

where

$$\delta_i(t) = \begin{cases} -\eta^{0.25} & \text{if } a_i(t) = \eta^{0.25} \\ \eta^{0.25} & \text{if } a_i(t) = -\eta^{0.25} \\ \sim \{-\eta^{0.25}, \eta^{0.25}\} & \text{if } a_i(t) = 0 \end{cases} \tag{32}$$

Besides, we use a new initialization for $\boldsymbol{a}$: for every $i \in [m]$, we have

$$a_i(0) = \begin{cases} -\eta^{0.25} & \text{with probability } 1/4 \\ \eta^{0.25} & \text{with probability } 1/4 \\ 0 & \text{with probability } 1/2 \end{cases} \tag{33}$$

Notice that $a_i(0)$ follows the stationary distribution. Since the transition matrix of $a_i$ is

$$Q = \begin{bmatrix} 0 & 1 & 0 \\ \frac{1}{2} & 0 & \frac{1}{2} \\ 0 & 1 & 0 \end{bmatrix}$$

It is easy to verify that $[1/4, 1/2, 1/4] = [1/4, 1/2, 1/4] \cdot Q$.

Thus by large number law, we can approximate $\|\boldsymbol{a}(t)\|^2$ by

$$\|\boldsymbol{a}(t)\|^2 = \sum_{i=1}^{m} |a_i(t)|^2 = m \cdot \mathbb{E}[|a_i(t)|^2] = \frac{m \cdot \eta^{0.5}}{2} \tag{34}$$

**Lemma J.3.** *Using algorithm 2, we have*

$$\mathbb{E}[\boldsymbol{\theta}(t)] = \mathbb{E}[\boldsymbol{\theta}(0)] \tag{35}$$

**Proof.** Notice that

$$\boldsymbol{W}(t+1) = \boldsymbol{W}(t+1) - \eta \cdot \boldsymbol{a}(t) \cdot \frac{1}{n} \sum_{i=1}^{n} (\boldsymbol{a}(t)^\top \boldsymbol{W}(t)\boldsymbol{x_i} - y_i) \cdot \boldsymbol{x_i}^\top \tag{36}$$

$$\boldsymbol{a}(t+1) = \boldsymbol{a}(t) + \boldsymbol{\delta}(t)^\top \tag{37}$$

Where $\boldsymbol{\delta}(t) = [\delta_1(t), \delta_2(t), \cdots, \delta_m(t)]^\top$.

Notice that $\boldsymbol{a}(t+1)^\top \cdot \boldsymbol{a}(t) = 0$, then we have

$$\boldsymbol{\theta}(t+1) = \boldsymbol{a}(t+1)^\top \cdot \boldsymbol{W}(t+1) \tag{38}$$

$$= (\boldsymbol{a}(t)^\top + \boldsymbol{\delta}(t)) \cdot (\boldsymbol{W}(t) - \eta \cdot \boldsymbol{a}(t) \cdot \frac{1}{n} \sum_{i=1}^{n} (\boldsymbol{a}(t)^\top \boldsymbol{W}(t)\boldsymbol{x}_i - y_i) \cdot \boldsymbol{x}_i^\top) \tag{39}$$

$$= \boldsymbol{\theta}(t) + \boldsymbol{\delta}(t) \cdot \boldsymbol{W}(t) - (\boldsymbol{a}(t+1)^\top \cdot \boldsymbol{a}(t)) \cdot \eta \cdot \frac{1}{n} \sum_{i=1}^{n} (\boldsymbol{a}(t)^\top \boldsymbol{W}(t)\boldsymbol{x}_i - y_i) \cdot \boldsymbol{x}_i^\top \tag{40}$$

$$= \boldsymbol{\theta}(t) + \boldsymbol{\delta}(t) \cdot \boldsymbol{W}(t) \tag{41}$$

Since $\mathbb{E}[\boldsymbol{\delta}(t)] = \boldsymbol{0}$, we have

$$\mathbb{E}[\boldsymbol{\theta}(t)] = \mathbb{E}[\boldsymbol{\theta}(0)]$$

$\square$

### J.3.1 PROGRESSIVE DIMINISHING OF NORM USING ALGORITHM 2

Since $\boldsymbol{x}_i \sim N(0, I)$ i.i.d, by law of large numbers, $\frac{1}{n} \sum_{i=1}^{n} \boldsymbol{x}_i \cdot \boldsymbol{x}_i^\top$ is approximately to $\mathbb{E}[\boldsymbol{x}_i \cdot \boldsymbol{x}_i^\top] = I$.
So we have

$$\boldsymbol{w_i}(t+1) = \boldsymbol{w_i}(t) - \eta \cdot a_i(t) \cdot (\boldsymbol{\theta}(t) - \boldsymbol{\theta}^*)^\top \tag{42}$$

$$a_i(t+1) = a_i(t) + \delta_i(t) \tag{43}$$

Square both sides of the equation and we have

$$\|\boldsymbol{w}_i(t+1)\|^2 = \|\boldsymbol{w}_i(t)\|^2 - 2\eta \cdot a_i(t) \cdot (\boldsymbol{\theta}(t) - \boldsymbol{\theta}^*)^\top \cdot \boldsymbol{w}_i(t+1) + \eta^2 \cdot a_i(t)^2 \cdot \|\boldsymbol{\theta}(t) - \boldsymbol{\theta}^*\|^2 \tag{44}$$

Since $|a_i(t)| \leq \eta^{0.25}$, we have

$$\frac{1}{m}\sum_{i=1}^{m}\|\boldsymbol{w}_i(t+1)\|^2 = \frac{1}{m}\sum_{i=1}^{m}\|\boldsymbol{w}_i(t)\|^2 - 2\eta \cdot (\boldsymbol{\theta}(t) - \boldsymbol{\theta}^*)^\top \cdot \frac{1}{m}\sum_{i=1}^{m}a_i(t)\boldsymbol{w}_i(t) + \eta^2 \cdot \frac{1}{m}\sum_{i=1}^{m}a_i(t)^2\|\boldsymbol{\theta}(t) - \boldsymbol{\theta}^*\|^2$$

(45)

$$= \frac{1}{m}\sum_{i=1}^{m}\|\boldsymbol{w}_i(t)\|^2 - 2\eta \cdot \frac{1}{m} \cdot (\boldsymbol{\theta}(t) - \boldsymbol{\theta}^*)^\top \cdot \boldsymbol{\theta}(t) + \eta^2 \cdot \frac{1}{m}\sum_{i=1}^{m}a_i(t)^2\|\boldsymbol{\theta}(t) - \boldsymbol{\theta}^*\|^2$$

(46)

$$\leq \frac{1}{m}\sum_{i=1}^{m}\|\boldsymbol{w}_i(t)\|^2 - 2\eta \cdot \frac{1}{m} \cdot (\boldsymbol{\theta}(t) - \boldsymbol{\theta}^*)^\top \cdot \boldsymbol{\theta}(t) + \eta^2 \cdot (\eta^{0.25})^2\|\boldsymbol{\theta}(t) - \boldsymbol{\theta}^*\|^2$$

(47)

$$= \frac{1}{m}\sum_{i=1}^{m}\|\boldsymbol{w}_i(t)\|^2 - (\frac{2\eta}{m} - \eta^{2.5})\|(\boldsymbol{\theta}(t) - \boldsymbol{\theta}^*)\|^2 - \frac{2\eta}{m} \cdot (\boldsymbol{\theta}(t) - \boldsymbol{\theta}^*)^\top \cdot \boldsymbol{\theta}^*$$

(48)

**Lemma J.4.** *If we set $a_i(0)$ to stationary distribution, then for any $t, k \in \mathbf{N}^+$, we have*

$$\mathbf{E}[\delta_i(t) \cdot \delta_i(t+k)] = \begin{cases} -\dfrac{\eta^{0.5}}{2} & \text{if } k = 1 \\ 0 & \text{if } k > 1 \end{cases}$$

(49)

**Proof.** Without loss of generality, assume we know that $\delta_i(t) = \eta^{0.25}$, then the distribution changes to $(0, \frac{1}{2}, \frac{1}{2})$. So we have

$$\mathbb{E}[\delta_i(t) \cdot \delta_i(t+1)] = \frac{1}{2} \cdot \eta^{0.25} \cdot (-\eta^{0.25}) + \frac{1}{2} \cdot 0 = -\frac{\eta^{0.5}}{2}$$

Then just after that, the distribution changes from $(0, \frac{1}{2}, \frac{1}{2})$ to stationary distribution $(\frac{1}{4}, \frac{1}{2}, \frac{1}{4})$ again. Thus for any $k > 1$ we have

$$\mathbb{E}[\delta_i(t) \cdot \delta_i(t+k)] = 0$$

$\square$

The following lemma reveals the process generated in Phase (1), thereby providing a proof for Theorem J.2.

**Lemma J.5** (Progressively diminishing under simulation setup)**.** *Under the setting in Section J.1 and assume KITINet is under thermal equilibrium, there exists a step $t_0 \leq \frac{1}{\eta^2}$ such that*

$$\mathbb{E}[\frac{1}{m}\sum_{i=1}^{m}\|\boldsymbol{w}_i(t_0)\|^2] \leq \sqrt{\eta}$$

(50)

**Proof.** Firstly, we give a lower bound of $\mathbb{E}[\|\boldsymbol{\theta}(t) - \boldsymbol{\theta}^*\|^2]$, which shows the decrease of each step. According to Equation (41), we retain

$$\boldsymbol{\theta}(t) - \boldsymbol{\theta}^* = \boldsymbol{\theta}(0) - \boldsymbol{\theta}^* + \sum_{i=0}^{t-1}\boldsymbol{\delta}(i) \cdot \boldsymbol{W}(i)$$

(51)

And Since $\mathbb{E}[\boldsymbol{\delta}(i)] = 0$, we have

$$\mathbb{E}[(\boldsymbol{\theta}(0) - \boldsymbol{\theta}^*) \cdot \sum_{i=0}^{t-1}\boldsymbol{\delta}(i) \cdot \boldsymbol{W}(i)] = \mathbb{E}[\boldsymbol{\delta}(i)] \cdot \mathbb{E}[(\boldsymbol{\theta}(0) - \boldsymbol{\theta}^*) \cdot \sum_{i=0}^{t-1}\boldsymbol{W}(i)] = 0$$

(52)

So we have

$$\mathbb{E}[\|(\boldsymbol{\theta}(t) - \boldsymbol{\theta}^*)\|^2] = \mathbb{E}[\|(\boldsymbol{\theta}(0) - \boldsymbol{\theta}^*)\|^2] + (\eta^{0.25})^2 \cdot \sum_{i=0}^{t-1}\mathbb{E}[\|\boldsymbol{W}(i)\|^2] + 2\sum_{i=0}^{t-1}\sum_{j<i}\mathbb{E}[(\boldsymbol{\delta}(i)\boldsymbol{W}(i)) \cdot (\boldsymbol{\delta}(j)\boldsymbol{W}(j))^\top]$$

(53)

Notice that

$$\mathbb{E}[(\boldsymbol{\delta}(i)\boldsymbol{W}(i)) \cdot (\boldsymbol{\delta}(i+1)\boldsymbol{W}(i+1)^\top] = \mathbb{E}[\boldsymbol{\delta}(i)(\boldsymbol{W}(i)\boldsymbol{W}(i+1)^\top)\boldsymbol{\delta}(i+1)^\top]$$

$$= \sum_{1 \le l, r \le m} \mathbb{E}[\delta_l(i)\delta_r(i+1) \cdot (\boldsymbol{W}(i)\boldsymbol{W}(i+1)^\top)[l,r]]$$

$$= \sum_{1 \le l, r \le m} \mathbb{E}[\delta_l(i)\delta_r(i+1) \cdot \boldsymbol{w}_l(i)^\top \boldsymbol{w}_r(i+1)]$$

Case 1: $r \ne l$. In this case, $\delta_l$ are independent with $\delta_r$. So we have

$$\mathbb{E}[\delta_l(i)\delta_r(i+1) \cdot \boldsymbol{w}_l(i)^\top \boldsymbol{w}_r(i+1)]$$

$$= \mathbb{E}[\delta_l(i)\delta_r(i+1) \cdot \boldsymbol{w}_l(i)^\top (\boldsymbol{w}_r(i) - \eta \cdot a_r(i) \cdot (\boldsymbol{\theta}(i) - \boldsymbol{\theta}^*)^\top]$$

$$= \mathbb{E}[\delta_l(i)\delta_r(i+1) \cdot \boldsymbol{w}_l(i)^\top \boldsymbol{w}_r(i)] - \mathbb{E}[\delta_l(i)\delta_r(i+1) \cdot \boldsymbol{w}_l(i)^\top \eta \cdot a_r(i) \cdot (\boldsymbol{\theta}(i) - \boldsymbol{\theta}^*)^\top]$$

$$= \mathbb{E}[\delta_l(i)] \cdot \mathbb{E}[\delta_r(i+1) \cdot \boldsymbol{w}_l(i)^\top \boldsymbol{w}_r(i)] - \mathbb{E}[\delta_l(i)] \cdot \mathbb{E}[\delta_r(i+1) \cdot \boldsymbol{w}_l(i)^\top \eta \cdot a_r(i) \cdot (\boldsymbol{\theta}(i) - \boldsymbol{\theta}^*)^\top]$$

$$= 0$$

Case 2: $r = l$, by Lemma J.4, we have

$$\mathbb{E}[\delta_l(i)\delta_l(i+1) \cdot \boldsymbol{w}_l(i)^\top \boldsymbol{w}_l(i+1)]$$

$$= \mathbb{E}[\delta_l(i)\delta_l(i+1) \cdot \boldsymbol{w}_l(i)^\top (\boldsymbol{w}_l(i) - \eta \cdot a_l(i) \cdot (\boldsymbol{\theta}(i) - \boldsymbol{\theta}^*)^\top)]$$

$$= \mathbb{E}[\delta_l(i)\delta_l(i+1) \cdot \boldsymbol{w}_l(i)^\top \boldsymbol{w}_l(i)] - \mathbb{E}[\delta_l(i)\delta_l(i+1) \cdot \boldsymbol{w}_l(i)^\top \eta \cdot a_l(i) \cdot (\boldsymbol{\theta}(i) - \boldsymbol{\theta}^*)^\top]$$

$$= \mathbb{E}[\delta_l(i)\delta_l(i+1)] \cdot \mathbb{E}[\|\boldsymbol{w}_l(i)\|^2] - \mathbb{E}[\delta_l(i)\delta_l(i+1) \cdot \boldsymbol{w}_l(i)^\top \eta \cdot a_l(i) \cdot (\boldsymbol{\theta}(i) - \boldsymbol{\theta}^*)^\top]$$

$$= -\frac{\eta^{0.5}}{2}\mathbb{E}[\|\boldsymbol{w}_l(i)\|^2] - \eta\mathbb{E}[\delta_l(i)\delta_l(i+1) \cdot a_l(i) \cdot (\boldsymbol{\theta}(i) - \boldsymbol{\theta}^*)^\top \cdot \boldsymbol{w}_l(i)]$$

$$= -\frac{\eta^{0.5}}{2}\mathbb{E}[\|\boldsymbol{w}_l(i)\|^2] - O(\eta^{1.75})$$

Thus we have

$$\mathbb{E}[(\boldsymbol{\delta}(i)\boldsymbol{W}(i)) \cdot (\boldsymbol{\delta}(i+1)\boldsymbol{W}(i+1))^\top] = \sum_{1 \le l, r \le m} \mathbb{E}[\delta_l(i)\delta_r(i+1) \cdot \boldsymbol{w}_l(i)^\top \boldsymbol{w}_r(i+1)]$$

$$= \sum_{l=1}^{m} \mathbb{E}[\delta_l(i)\delta_l(i+1)] \cdot \boldsymbol{w}_l(i)^\top \boldsymbol{w}_l(i+1))$$

$$= -\frac{\eta^{0.5}}{2} \sum_{l=1}^{m} \mathbb{E}[\|\boldsymbol{w}_l(i)\|^2] - O(m \cdot \eta^{1.75})$$

$$= -\frac{\eta^{0.5}}{2}\mathbb{E}[\|\boldsymbol{W}(i)\|^2] - O(m \cdot \eta^{1.75})$$

Then we retain

$$\sum_{i=0}^{t-1} \sum_{j<i} \mathbb{E}[(\boldsymbol{\delta}(i)\boldsymbol{W}(i)) \cdot (\boldsymbol{\delta}(j)\boldsymbol{W}(j))^\top] = \sum_{i=0}^{t-2} \mathbb{E}[(\boldsymbol{\delta}(i)\boldsymbol{W}(i)) \cdot (\boldsymbol{\delta}(i+1)\boldsymbol{W}_{i+1})^\top] \quad (54)$$

$$= -\frac{\eta^{0.5}}{2} \cdot \sum_{i=0}^{t-2} \mathbb{E}[\|\boldsymbol{W}(i)\|^2] - O(m \cdot \eta^{1.75}) \quad (55)$$

Combining Equation (53) and Equation (55), we have

$$\mathbb{E}[\|(\boldsymbol{\theta}(t) - \boldsymbol{\theta}^*)\|^2] = \mathbb{E}[\|(\boldsymbol{\theta}(0) - \boldsymbol{\theta}^*)\|^2]$$

$$+ \eta^{0.5} \cdot \sum_{i=0}^{t-1} \mathbb{E}[\|\boldsymbol{W}(i)\|^2] + 2 \sum_{i=0}^{t-1} \sum_{j<i} \mathbb{E}[(\boldsymbol{\delta}(i)\boldsymbol{W}(i)) \cdot (\boldsymbol{\delta}(j)\boldsymbol{W}(j))^\top] + O(m \cdot \eta^{1.75})$$

$$= \mathbb{E}[\|(\boldsymbol{\theta}(0) - \boldsymbol{\theta}^*)\|^2] + \eta^{0.5} \cdot \sum_{i=0}^{t-1} \mathbb{E}[\|\boldsymbol{W}(i)\|^2] - \eta^{0.5} \cdot \sum_{i=0}^{t-2} \mathbb{E}[\|\boldsymbol{W}(i)\|^2] + O(m \cdot \eta^{1.75})$$

$$= \mathbb{E}[\|(\boldsymbol{\theta}(0) - \boldsymbol{\theta}^*)\|^2] + \eta^{0.5} \mathbb{E}[\|\boldsymbol{W}(t-1)\|^2] - O(m \cdot \eta^{1.75})$$

$$\geq \mathbb{E}[\|\boldsymbol{\theta}^\star\|^2] + \eta^{0.5} \mathbb{E}[\|\boldsymbol{W}(t-1)\|^2]$$

Then we give a proof of this theorem by contradiction. Assume for every step $t \leq \frac{1}{\eta^2}$, we have

$$\mathbb{E}[\frac{1}{m} \sum_{i=1}^{m} \|\boldsymbol{w_i}(T)\|^2] \geq \sqrt{\eta} \tag{56}$$

i.e.

$$\mathbb{E}[\|\boldsymbol{W}(t)\|^2] \geq m \cdot \sqrt{\eta} \tag{57}$$

When $T = \dfrac{1}{\eta^2}$, according to Equation (48), sum up from $0$ to $T-1$ and we have

$$\mathbb{E}[\frac{1}{m} \sum_{i=1}^{m} \|\boldsymbol{w_i}(T)\|^2] = \mathbb{E}[\frac{1}{m} \sum_{i=1}^{m} \|\boldsymbol{w}_i(0)\|^2] - (\frac{2\eta}{m} - \eta^{2.5}) \cdot \sum_{t=0}^{T-1} \mathbb{E}[\|(\boldsymbol{\theta}(t) - \boldsymbol{\theta}^*)\|^2] - \frac{2\eta}{m} \cdot \sum_{t=0}^{T-1} \mathbb{E}[(\boldsymbol{\theta}(t) - \boldsymbol{\theta}^*)^\top \cdot \boldsymbol{\theta}^*]$$

$$\leq \mathbb{E}[\frac{1}{m} \sum_{i=1}^{m} \|\boldsymbol{w}_i(0)\|^2] - (\frac{2\eta}{m} - \eta^{2.5}) \cdot (T \cdot \mathbb{E}[\|\boldsymbol{\theta}^*\|^2] + \eta^{0.5} \sum_{t=0}^{T-1} \mathbb{E}[\|\boldsymbol{W}_t\|^2]) + \frac{T \cdot 2\eta}{m} \cdot \mathbb{E}[\|\boldsymbol{\theta}^*\|^2]$$

$$\leq \mathbb{E}[\frac{1}{m} \sum_{i=1}^{m} \|\boldsymbol{w}_i(0)\|^2] - 2\eta^{1.5} \cdot T \cdot \sqrt{\eta} + O(T \cdot \eta^{2.5})$$

$$= 1 - 2\eta^{1.5} \cdot \frac{1}{\eta^2} \cdot \sqrt{\eta} + O(T \cdot \eta^{2.5}) = -1 + O(T \cdot \eta^{2.5}) < 0$$

Which is absolutely a contradiction! Therefore, we complete the proof. $\square$

