# OpenReview forum: "KITINet: Kinetics Theory Inspired Network Architectures with PDE Simulation Approaches"
_ICLR.cc/2026/Conference — ICLR 2026 Conference Withdrawn Submission_

### Official Review · Reviewer_mfJA · 2025-10-31

**Soundness:** 2
**Presentation:** 2
**Contribution:** 2
**Rating:** 4
**Confidence:** 3

**Summary:**

The authors propose a new way of combining the residual stream and processed activations in residual layers. Inspired by Neural ODEs, they consider the residual stream as position $x$ and the processed activations as velocity $v$ for which they reshape the hidden feature dimension into a particle simulation that they simulate for one time step. This mechanism is only active during training. It falls back to simple residual-stream addition during inference. The authors argue that their method improves parameter condensation since particle collisions cause spatial distributions to be more dispersed. Since residual layers are common in most modern deep learning architectures, their extension can be monkey-patched into various networks for different tasks. The authors demonstrate this with GPT-like next-token prediction, ResNet-style image classification, PDE operator learning, and BERT-like text classification, for which they report improvements over the baseline.

**Strengths:**

- The authors make an interesting connection to kinetic gas theory.
- They use a wide range of experiments and can consistently demonstrate an improvement over the chosen baseline (even if it might not hold in the strictest statistical setting).
- The modification they suggest seems plug-and-play and should allow for easy integration into existing architectures. While it does influence training time, at inference the network behaves as if it has a classical residual connection.

**Weaknesses:**

- The authors motivate their method by claiming a certain "feature-space distance". It is unclear to me how they define this distance and why they end up with the value $L \approx 3.29$. Moreover, I encourage the authors to add a reference for the valid region of the BTE.
- Some results are not statistically significant. For example, in Table 1 most values when comparing between GPT2 and KITTI-GPT2 are within +-1.96 standard error range. Considering that there are no computational advantages (The authors note "KITTINet consistently reaches target  accuracy levels with approximately 20% fewer training steps than the baseline, which covers the overheads of its additional computation during training."), there seems to be little incentive to use KITTINet over the baseline method
- The entire section 2 seems a bit unclear to me. The connection between kinetic gas theory and residual layers requires a stronger motivation. Moreover, this section breaks with the natural flow of the paper since it foreshadows quantities that are only introduced later (e.g., in line 85)

**Questions:**

* Please fix the citations in the references to datasets, e.g., in line 308
* The integral of Eq. (7) is missing the integrand $dt$
* In lines 401, the authors perform an ablation on varying the number of `collision_heads` for the FNO. Since FNOs typically use hidden feature sizes of $O(100)$, I wonder how the authors distribute this into $2^{10} \approx 1000$ heads?

---

### Official Review · Reviewer_QJdT · 2025-10-31

**Soundness:** 3
**Presentation:** 3
**Contribution:** 3
**Rating:** 4
**Confidence:** 3

**Summary:**

This paper examines residual learning through the perspective of kinetic theory, non-equilibrium particle dynamics. existing residual modules are designed heuristically, neglecting the rich dynamics of particle interaction or energy exchange. existing dynamical systems perspective models residual networks as discretized odes, ignoring stochastic interaction.

The paper proposes a new residual module that models feature updates as evolution of a particle system, mimicking collisions and enabling information propagation through physical interaction. The evolution is simulated numerically where each channel is a particle whose interaction is simulated by a discretized PDE solver.

Analysis shows that the new mechanism is an implicit regularization approach. Experiments in multiple settings including language modeling, image classification, scientific computing show consistent improvement.

**Strengths:**

The proposed module is novel with motivating theory, replacing the residual connection operation of addition as particle collision dynamics.

The writing is of good quality.

Experiments are extensive and show improvement on a number of benchmarks. The experiments range of large language model pretraining to image classification to neural operator learning.

Parameter condensation is promoted by the method which may be an explanation of generalization

**Weaknesses:**

Motivation for modeling the interaction as particles is not clear.

There is no discussion of related work in the main paper. There is some discussion in the appendix that should be in the main part.

Parts of the paper are confusing and not clearly written (say, lines 158-160). The preliminary section appears to have details not later used.

The description of the architectural components is also confusing. The description of the simulation in equations 2-5 is opaque and hard to understand.

Modeling collisions requires numerical solution of a PDE which may be more expensive.

I am not  familiar with parameter condensation. It would be useful to have a description of this phenomenon.

**Questions:**

I am missing some part of the motivation. Why would kinetic theory and stochastic particle collision dynamics be a good model for residual connections in neural networks?

Furthermore, in the introduction, what is the relation of physics inspired architectures to residual networks. As far as I can tell these are two unrelated things where physics inspired architectures embed physical principles into architectures which may or may not employ residual modules.

In line 44-45 it is said that dynamical systems as ODEs fails to account for stochastic, collision driven interaction. Is this kind of interaction present in residual networks? Have the authors (or prior work) shown this behavior in NNs that the ODE perspective fails to account for?

This leads to the question of whether this work is an explanation of residual modules? or a new physics-inspired architecture based on particle collisions?

I understand that to model collisions the Boltzmann transport equation needs to be solved. How efficient is this? How does this effect the time complexity of the models in relation to the original models in the experiments?

Table 4 compares with FNO in terms of root MSE. It is difficult to judge the raw MSE scores. What are the results in normalized error?

line 61: What is the motivation for physically-grounded architectures for language or text classification?
line 65: ‘heated phenomenon’
In line 86 I do not understand what batchnorm has to do with kinetic theory or particle dynamics

---

### Official Review · Reviewer_V9j8 · 2025-11-01

**Soundness:** 2
**Presentation:** 2
**Contribution:** 3
**Rating:** 4
**Confidence:** 2

**Summary:**

This paper introduces KITINet, a novel, parameter-free module to replace the standard residual connection. The core idea is to reinterpret feature propagation through the lens of kinetic theory, treating feature channels as "particles" (with positions $x$ and velocities $v$). The module simulates the stochastic evolution of this particle system using a "DSMC-inspired" discretized solver for the Boltzmann transport equation (BTE). This physics-based collision simulation is active only during training and acts as an implicit regularizer. The authors claim this method induces "network parameter condensation," where parameters consolidate into a sparse, dominant subset of channels. The module is turned off during inference, where it reverts to a standard residual addition, thus incurring zero additional parameters or computational cost at inference time. The authors demonstrate consistent performance improvements across a wide range of tasks.

**Strengths:**

- The conceptual framework seems original. Linking residual learning to non-equilibrium particle dynamics and the Boltzmann transport equation is a novel approach.
- The proposed module is a training-time-only regularizer that is free at deployment. This is a desirable trade-off, as it allows for a more robust and better-generalized model with no extra-test time computation.
- Testing suite seems to show that the approach is quite general: the experiments span diverse domains including language modeling, image classification and PDE solving. The improvements on the PDE front seem rather substantial.

**Weaknesses:**

- Although the paper mentions that KITINet reaches target accuracy "approximately 20% fewer training steps," the actual computational overhead during training is not thoroughly analyzed. Would like to see some more training details and logs. Might be helpful to see more results on the FLOPs / training time per epoch / iteration. If we were to plot out the training FLOPs against the accuracy, can we expect to see the method beating current baselines?
- The improvements on general-purpose benchmarks like LLMs and image classification seem rather modest. However, I'm not strongly familiar with benchmarks in that domain. On the front of PDEs, the selection of benchmarks could be strengthened. To fully evaluate the method's efficacy against the current state-of-the-art and ensure fair comparisons, it would be beneficial to test it on more recent, standardized benchmarks. The datasets from 'TheWell', for example, would be an excellent candidate for this.
- The paper does not investigate why KITINet works better in some contexts than others. Does the "particle collision" metaphor universally helpful, or is it (as the PDE results suggest) particularly effective for problems that are already governed by diffusion or particle-like dynamics?
- Does this module benefit all architectures equally? The paper applies it to both FNO and OFormer. A direct comparison is needed. Does the collision mechanism provide more of an advantage to the Transformer's attention mechanism or to the FNO's global convolution?

**Questions:**

Please address the weaknesses

---

### Official Review · Reviewer_gGkH · 2025-11-03

**Soundness:** 2
**Presentation:** 1
**Contribution:** 2
**Rating:** 2
**Confidence:** 3

**Summary:**

This paper considers a modification to the standard residual layer by incorporating a new stochastic connection inspired by molecular kinetics with collision. The residual connection is a simplified version of a Monte-Carlo-based technique used for Boltzmann Transport equations. By replacing the residual connections in many architectures in different domains, the authors claim improved performance. The article also highlights the notion of "network parameter condensation" as a metric that is used to claim the improvement.

**Strengths:**

The idea is interesting, the notion of network parameter condensation appears to be a new metric to consider. The approach of modifying the residual connection using the Direct Simulation Monte Carlo (DSMC) time-step appears widely applicable to generic tasks using residual connections.

**Weaknesses:**

- It is not clear in the manuscript why the "network parameter condensation" is a useful measure of performance.
- The authors say that the design principles are largely heuristic, but appears to simply consider the "a different residual connections" in a modular fashion applied to existing architectures.
- The "residual connection" originally is a very simple operation, whereas the proposed new residual connection involves new parameters in the "residual layer" $l_\theta$. This simply makes the terminology confusing because then this layer is no longer a "residual layer"? For example, you can add a residual layer on top of their "DSMC-inspired module" and calling this new module a residual connection simply causes confusion.
- It is also not clear to this reviewer what Theorem 4.1 is attempting to achieve, the terms used in the statements were not introduced properly.
- The improvements against existing models are very modest, and does not appear significant.

**Questions:**

- Regarding the notion "network parameter condensation": Is there a way to connect the sparsity of the features to the performance of the task? Is this connection valid both for LLM-based tasks and Operator Learning tasks?
- On the design principles: How is the design principle different than a ODE-based view? In a way the Boltzmann Transport Equation is being evolved in time and the proposed methods can be viewed as an ODE-based approach as well.
- Regarding Theorem 4.1: What is this assumption that a neural network is "under thermal equilibrium" and what does the "rapid convergence process" mean here.

**Details Of Ethics Concerns:**

- Is there a way to connect the sparsity of the features to the performance of the task? Is this connection valid both for LLM-based tasks and Operator Learning tasks?

---

### Note · Authors · 2025-11-26

I have read and agree with the venue's withdrawal policy on behalf of myself and my co-authors.